**Advantages of assimilating multi-spectral satellite retrievals of atmospheric composition: A demonstration using MOPITT CO products**

**Wenfu Tang[1], Benjamin Gaubert[1], Louisa K. Emmons[1], Daniel Ziskin[1], Debbie Mao[1], David P. Edwards[1], Avelino F. Arellano[2], Kevin Raeder[3], Jeffrey L. Anderson[3], and Helen M. Worden[1]**

[1]Atmospheric Chemistry Observations & Modeling Laboratory, National Center for Atmospheric Research, Boulder, CO, USA

[2]Dept. of Hydrology and Atmospheric Sciences, University of Arizona, Tucson, AZ, USA

[3]Computational and Information Systems Laboratory, National Center for Atmospheric Research, Boulder, CO, USA

Correspondence: Wenfu Tang (wenfut@ucar.edu)

**Abstract**

The Measurements Of Pollution In The Troposphere (MOPITT) is an ideal instrument to understand the impact of (1) assimilating multispectral/joint retrievals versus single-spectral products, (2) assimilating satellite profile products versus column products, and (3) assimilating multispectral/joint retrievals versus assimilating individual products separately. We use the Community Atmosphere Model with chemistry with the Data Assimilation Research Testbed (CAM-chem+DART) to assimilate different MOPITT CO products to address these three questions. Both anthropogenic and fire CO emissions are optimized in the data assimilation experiments. The results are compared with independent CO observations from TROPOspheric Monitoring Instrument (TROPOMI), the Total Carbon Column Observing Network (TCCON), NOAA Carbon Cycle Greenhouse Gases (CCGG) sites, In-service Aircraft for a Global Observing System (IAGOS), and Western wildfire Experiment for Cloud chemistry, Aerosol absorption and Nitrogen (WE-CAN). We find that (1) assimilating the MOPITT joint (multispectral Near-IR and Thermal-IR) column product leads to better model-observation agreement at and near the surface than assimilating the MOPITT Thermal-IR-only column retrieval. (2) Assimilating column products has a larger impact and improvement for background and large-scale CO compared to assimilating profile products due to vertical localization in profile assimilation. However, profile assimilation can out-perform column assimilations in fire-impacted regions and near the surface. (3) Assimilating multispectral/joint products results in similar or slightly better agreement with observations compared to assimilating the single-spectral products separately.

**1 Introduction**

With the increasing availability of satellite remote sensing instruments measuring atmospheric composition, there is potential to produce multispectral retrievals of several species, making use of thermal-infrared (TIR) and near-infrared (NIR) radiances from collocated instruments on the same satellite such as IASI (Infrared Atmospheric Sounding Interferometer) and GOME-2 (Global Ozone Monitoring Experiment-2) on the European MetOp satellites (Cuesta et al., 2013), or flying in close formation, such as on the NASA A-train and the NOAA's JPSS (Joint Polar Satellite System), e.g., OMI (Ozone Monitoring Instrument, Levelt et al., 2018), AIRS

(Atmospheric Infrared Sounder, Fu et al., 2018), OMPS (Ozone Mapping and Profiler Suite, Flynn et al., 2014), TROPOspheric Monitoring Instrument (TROPOMI, Veefkind et al., 2012) and CrIS (Cross-track Infrared Sounder, Fu et al., 2016). TIR retrievals use thermal contrast while NIR retrievals use reflected solar radiance from the surface. Taking MOPITT as an example, the TIR retrieval can provide vertical profiles with limited sensitivity to the surface while the NIR retrieval only provide total column product with some sensitivity to the surface (Figure 1).

The multispectral products have shown considerable increases in the vertical sensitivity of the retrievals for lowermost tropospheric ozone ($O_3$) (e.g., Worden et al., 2007; Natraj et al., 2011; Fu 2018), carbon monoxide (CO) (Worden et al., 2010; Fu et al., 2016) and methane ($CH_4$) (Schneider et al. 2022). Multispectral retrievals could be made using the co-located overpass made by low earth orbit and geostationary satellite such as, e.g., Geostationary Interferometric Infrared Sounder (GIIRS, Zeng et al., 2023), Geostationary Environment Monitoring Spectrometer (GEMS, Kim et al., 2020), Geostationary Extended Observations (GeoXO; Kopacz et al., 2023) and Tropospheric emissions: Monitoring of pollution (TEMPO, Chance et al., 2019). Table 1 shows the developed and potential multispectral products. It is important to understand the value of assimilating a multispectral product versus assimilating a single-spectral range product, and the value of assimilating a multispectral product versus separately assimilating single-spectral range products that are used to retrieve the multispectral products.

**Table 1**. Developed and potential multispectral satellite retrievals. Shown in the table are satellites, their NIR and/or TIR spectral ranges (in µm), and potential chemical species from the multispectral retrievals.

| Morning Overpass | Afternoon Overpass | Geostationary |
|---|---|---|
| MOPITT (2.3 & 4.7) <br><br> (CO) | AIRS (3.75–15.4) + OMI (0.27–0.5) <br><br> (O3) | GIIRS (East Asia) (0.55–14.2) + TROPOMI (2.3–2.4) <br> (CO, O3) |
| IASI (3.6–15.5) + GOME2 (0.24–0.79) <br> (O3) | TES (8.7–10.5) + OMI (0.27–0.5) <br><br> (O3) | GEMS (East Asia) (0.3–0.5) + IASI (3.6–15.5) <br> (O3) |
| | GOSAT (0.75–15) + TES (8.7–10.5) <br><br> (O3) | GEMS (East Asia) (0.3–0.5) + CrIS (3.9–15.4) <br> (O3) |
| | CrIS (3.9–15.4) + GOSAT-2 (0.3–14.3) <br><br> (CO, CH4) | TEMPO (N. America) (0.29–0.74) + IASI (3.6–15.5) <br> (O3) |
| | CrIS (3.9–15.4) + TROPOMI (2.3–2.4) <br><br> (CO, O3, CH4) | TEMPO (N. America) (0.29–0.74) + CrIS (3.9–15.4) <br> (O3) |

Total column observations of $O_3$, CO and Nitrogen Dioxide ($NO_2$) are now routinely assimilated in operational centers such as in the European Copernicus Atmosphere Monitoring Service (CAMS) program at the European Centre for Medium-Range Weather Forecasts (Inness et al., 2019; 2022) In addition, recently launched geostationary satellites such as GEMS and TEMPO will provide column products at high temporal resolution. While the satellite profile products are in general considered to contain more vertical information, it is important to understand the impacts of assimilating column products versus assimilating profile products and to understand what information is potentially missed by only assimilating column products. For example, Jiang et al. (2017) compared emission updates following the assimilation of the

Measurements of Pollution in the Troposphere (MOPITT) lowermost surface profile, the
tropospheric profile or the columns and identified errors indicative of model transport error
impacts on emission estimates.
The MOPITT instrument onboard the NASA Terra satellite is an ideal instrument to
address these three questions. MOPITT retrieves total column amounts and vertical profiles of CO
using both thermal-infrared (TIR) and near-infrared (NIR) measurements. In addition, MOPITT
also provides the multispectral TIR-NIR joint product, which has enhanced the sensitivity to near-
surface CO (Deeter et al., 2011, 2013; Worden et al., 2010). By comparing the results of
assimilating different combinations of MOPITT CO products, we will be able to address these two
questions.
To conduct the data assimilation experiments, we use the Community Atmosphere Model
with chemistry and the Data Assimilation Research Testbed (Anderson et al., 2009). CAM-
chem+DART has been previously used to assimilate MOPITT profile products (Arellano et al.,
2007; Barré et al., 2015; Gaubert et al., 2016, 2017, 2020, 2023). Here we present the first
assimilation of MOPITT column products within CAM-chem+DART. This new capability also
allows us to assimilate other satellite column products of CO and other chemical species in the
future. Anthropogenic and fire emissions are optimized separately in the data assimilation
experiments.
This paper aims to understand the impacts of (1) assimilating multispectral/joint products
versus single-spectral products, (2) assimilating satellite profile products versus column products,
and (3) assimilating multispectral/joint products versus assimilating individual products
separately. The paper is organized as follows: Section 2 describes CAM-chem, DART, and
methods, Section 3 describes datasets used for results evaluation, Section 4 presents data
assimilation diagnostics, Section 5 shows comparisons between data assimilation results and
independent observations, Section 6 discuss optimized emissions and CAM-chem simulations
with updated emissions, Section 7 is discussion and Section 8 concludes the study.
**Section 2: Methods and data**
**2.1 MOPITT products**
The Measurements of Pollution in the Troposphere (MOPITT) instrument on board the
NASA Terra satellite provides both thermal-infrared (TIR) and near-infrared (NIR) radiance
measurements since March 2000 (Deeter et al., 2003). CO total column amounts and volume
mixing ratio (VMR) profiles (10 vertical layers) are retrieved from the radiance measurements.
TIR is used to retrieve MOPITT TIR CO total column product and MOPITT TIR CO vertical
profile product; NIR is used to retrieve MOPITT NIR CO column product. Besides the TIR-only
and NIR-only products, multispectral (JNT) products are also provided by MOPITT by jointly
retrieving from TIR and NIR. JNT retrievals provide both MOPITT JNT CO total column product
and MOPITT JNT CO vertical profile product. JNT products have enhanced the sensitivity to near-
surface CO (Deeter et al., 2011, 2013; Worden et al., 2010). MOPITT products can be accessed
through https://search.earthdata.nasa.gov/search. In this study, we assimilate daytime MOPITT
version 9 products (Deeter et al., 2022) of TIR profile, TIR column, NIR column, JNT profile, and
JNT column in our experiments.
We use the error-weighted average of the MOPITT data within 1°×1° model grid and 6-
hourly bin (i.e., super-observations). Averaged daily numbers of daytime total super-observations

from MOPITT TIR, NIR, and JNT products during July 16[th] 2018 to August 14[th] 2018 is shown in Figure 2. The NIR product only covers the land while TIR and JNT products cover the land and ocean. Over the ocean, the JNT product is the same as the TIR product (Worden et al., 2010).

Data assimilation requires observation errors associated with the quantity assimilated. MOPITT provides 3 types of uncertainties/errors: total error, measurement error, and smoothing error in the products. Total error includes both measurement error and smoothing error. Since our observation operators include the smoothing by the MOPITT averaging kernels and the prior profiles, we only use the measurement error rather than total error provided by MOPITT for both column and profile products as smoothing error is already addressed by observation operators in the system (Rodgers, 2000). Specifically, for MOPITT profile products, measurement error is provided by the variable "MeasurementErrorCovarianceMatrix" while for MOPITT column products, measurement error is provided by the variable second column of the "RetrievedCOTotalColumnDiagnosticsDay".

## 2.2 CAM-chem

The Community Earth System Model (CESM) is a global Earth system model that includes the atmosphere, land, ocean, and ice components (Danabasoglu et al., 2020). CAM-chem (Emmons et al., 2020; Tilmes et al., 2019) is a global chemistry-climate model as a configuration of CESM version 2.2 (https://www2.acom.ucar.edu/gcm/cam-chem). CAM-chem accounts for physical, chemical and dynamical processes with a spatial resolution of 1.25° in longitude and 0.95° in latitude and 32 vertical layers with ~8 layers in boundary layer and ~10 layers in the free troposphere (Tang et al., 2023). We use the default MOZART-TS1 chemical mechanism, which includes comprehensive tropospheric and stratospheric chemistry with ~220 chemical species and 528 reactions (Emmons et al., 2020). The aerosol scheme used is the four-mode version of the Modal Aerosol Module (MAM4; Liu et al., 2016).

We use CAMS-GLOB-ANT v5.1 inventory (Soulie et al., 2023) for anthropogenic emissions and FINNv2.4 (Wiedinmyer et al., 2023) for fire emissions. CAMS-GLOB-ANT v5.1 provide monthly emissions and we generated daily files from the interpolation of the monthly values. The FINNv2.4 inventory provide daily fire emissions and are used directly. We update CO emission input files using the relative surface flux increments at every MOPITT CO assimilation step (6-hourly).

## 2.3 DART

DART is an open-source community facility for efficient ensemble data assimilation (https://dart.ucar.edu/). It is developed and maintained at the National Center for Atmospheric Research (NCAR). DART has been coupled with Community Atmosphere Model (CAM) for global meteorological data assimilation (CAM+DART; Raeder et al., 2012, 2021). Based on CAM+DART, the capability of chemical data assimilation using CAM-chem online chemistry and DART is developed and applied for scientific research (CAM-chem+DART; Arellano et al., 2007; Barré et al., 2015; Gaubert et al., 2016, 2017, 2020). Here, we use the Ensemble Adjustment Kalman Filter approach (EAKF; Anderson, 2001, 2003). The forecast ensemble is generated by 30 CAM-chem simulations with different initial conditions and emissions. The assimilation is performed using DART and produces an ensemble of optimized initial conditions and emissions, as described in Gaubert et al. (2023). Specifically, the state vector includes CO initial conditions, and CO emission fluxes that are ascribed to fires and anthropogenic sources. We use ensemble

mean at the forecast and the analysis step in the result sections. Ensemble mean of forecast is denoted by

$$\overline{x^f} = \frac{1}{N}\sum_{j=1}^{N} x_j^f \tag{1}$$

where $\overline{x^f}$ is the ensemble mean of "forecast", N is the ensemble size and $x_j^f$ is the forecast value of the j-th ensemble member. In our runs, DART uses EAKF, a deterministic ensemble square root filter for the analysis step. Unless noted otherwise, our setup is the same as in Gaubert et al., (2023). We slightly change the emission update to include a correction to the previous day (t-1) in order to smooth the emissions increments. Briefly, we apply multiplicative covariance inflation to the forecast ensemble before each analysis step to adjust the total error (model and observations) using the given observation error as reference (Anderson, 2007, 2009). The inflation parameter is also sequentially updated (Gharamti 2018) and varies in both space and time. Localization is commonly used in ensemble-based data assimilation to address insufficient ensemble sample size. Since the correlation is expected to decrease as separation increases, it empirically reduces the impact of an observation on model state variable as a function of distance using the Gaspari–Cohn localization function (Gaspari and Cohn, 1999). The spatial localization horizontal half width is 600 km and the vertical half width is 1200 m. The main difference between the profile and the column assimilation resides in the vertical localization. For each MOPITT retrieval, profile products have multiple observations at different layers but their impacts are vertically localized around 100 hPa. Therefore, not all vertical layers will be impacted. For the column data assimilation, there is no vertical localization in the column data assimilation except that the stratospheric (top 5) levels are not updated, as in the CO profile and meteorological DA. All vertical levels will be impacted by a single column value. In this case, if the mismatch is due to an underestimation of surface emissions rather than weak vertical transport, updating the upper tropospheric CO might lead to erroneous adjustments in CO abundance.

Forward operators (denoted as $H$ in DA terminology) are applied to project model field to observation space (i.e., expected observations). We use the forward operators introduced in Barré et al., (2015), consisting of i) estimating the log of a pressure weighted partial column volume mixing ratio that corresponds to the MOPITT grid and ii) applying the MOPITT averaging kernel and prior information as mentioned in section 2.1. In this study, we introduce an observation operator to assimilate the MOPITT columns in DART. That is, we estimate the retrieved column C (molecules cm$^{-2}$), using the MOPITT prior column $C_a$ and following Equation 3 of the MOPITT Version 9 Product User's Guide:

$$C = C_a + a(x_{CAM-chem} - x_a) \tag{2}$$

where $x_{CAM-chem}$ and $x_a$ are the modelled and the MOPITT a priori profiles expressed as $\log_{10}$(VMR) and $a$ is the total column averaging kernel. In this study, we assimilate both MOPITT profile and column products and compare the results.

**2.4 Data assimilation experiments setup**
There are 6 CAM-chem+DART runs (Figure 3). The first run is the spin-up/control run that starts on July 1$^{st}$ 2018. The spin-up/control run only assimilates meteorological observations and the state vector consists in wind, temperature, specific humidity, and surface pressure. Besides the spin-up/control run, there are 5 experiment runs that assimilate different MOPITT CO product(s) to update model CO. Note that the experiment runs not only assimilate MOPITT CO

products but also meteorological variables as in the spin-up/control run. The chemical state vector
(CO and CO emissions) and the meteorological state vector do not impact each other. However,
the updated meteorology due to meteorological data assimilation will impact the transport and
possibly chemistry of CO during the forecast step. The 5 experiment runs are:
(1) Column JNT assimilation (Exp1-CJ);
(2) Profile JNT assimilation (Exp2-PJ);
(3) Column TIR assimilation (Exp3-CT);
(4) Column TIR and Column NIR assimilation (Exp4-CT+CN);
(5) Profile TIR and Column NIR assimilation (Exp5-PT+CN).
These 5 experiment runs are designed to address a few scientific questions:
● The comparisons of Exp1-CJ and Exp2-PJ will show the impacts of the assimilation of
satellite profile versus column products.
● The comparisons of Exp1-CJ and Exp3-CT will show the difference caused by TIR-only
product versus joint product.
● The comparisons of Exp1-CJ and Exp4-CT+CN will show the impacts of assimilating joint
products (TIR+NIR) versus assimilating them separately for column products.
● The comparisons of Exp2-PJ and Exp5-PT+CN will show the impacts of assimilating joint
products (TIR+NIR) versus assimilating them separately for profile products.
The experiment runs start on July 16th 2018 and are initialized with the spin-up/control run.
Each experiment runs for 35 days considering the cost and constrain of computational allocation.
The first 20 days (July 11th to July 15th, 2018) are CO spin-up and the last 15 days (July 31st to
August 14th, 2018) are used for result analyses. The 15-day period are selected based on the spin-
up time – as shown by fractions of observations rejected by the assimilation system (Figure 4).
Quality checks are common in data assimilation as the algorithms are employed operationally for
near real time forecasting. We use the standard option in DART to do such quality checks. The
absolute value of the difference between the observed value and the prior ensemble mean estimate
is divided by the expected value of this difference. That expected value is the square root of the
sum of the specified observation error variance and the prior ensemble variance. If this ratio is
greater than a threshold, the observation is not used. The threshold ratio used here is three which
is commonly used for large tropospheric applications in DART (e.g., Gaubert et al., 2023).
Systematic errors are larger at the beginning of the spin-up, explaining the higher rejection rate.
As the assimilation proceeds and the forecast bias is reduced, the rejection rate goes down. The
experiments finished spinning up around 31 July. Each CAM-chem+DART run includes 30
ensemble members. These 30 ensemble members have different initial conditions and emissions
to represent model uncertainties. The analysis step is done every 6 hours. Anthropogenic and fire
emissions are optimized separately on a daily basis following the method described in Gaubert et
al. (2020, 2023).
**2.5 CAM-chem simulations with updated emissions**
To evaluate the updated emissions from the DA experiments, we conduct CAM-chem
simulations for the same period using the ensemble mean of the updated fire and anthropogenic
emissions. Hourly output is used for these simulations. Specifically, we conduct 6 CAM-chem
simulations:
(S1) Simulation with emissions from Exp1-CJ;
(S2) Simulation with emissions from Exp2-PJ;
(S3) Simulation with emissions from Exp3-CT;

(S4) Simulation with emissions from Exp4-CT+CN;
(S5) Simulation with emissions from Exp5-PT+CN;
(SControl) Simulation with original CAMS and FINN emissions.
**3 Datasets used for results evaluation**
**3.1 TROPOspheric Monitoring Instrument (TROPOMI)**
We use CO column retrieved from the TROPOMI instrument onboard the ESA's Sentinel-
5 Precursor (Veefkind et al., 2012) to evaluate model results. The spatial resolution of CO
retrievals is ~5.5 km × 7 km (Veefkind et al., 2012; Borsdorff et al., 2018). TROPOMI CO data
can be downloaded from https://s5phub.copernicus.eu/dhus/#/home. The TROPOMI Level 2 CO
(Apituley et al., 2018) is used here. The TROPOMI data are filtered following Landgraf et al.
(2018). To compare the model results with TROPOMI CO, we interpolate model outputs spatially
and temporally to match the locations and times of TROPOMI CO retrievals, and then apply
TROPOMI CO total column averaging kernels to the interpolated model CO profiles to obtain
modeled total CO columns (Apituley et al., 2018). TROPOMI CO data were compared to MOPITT
CO in Martínez-Alonso et al., (2020). TROPOMI and MOPITT data show good agreement in
terms of temporal and spatial patterns with global average biases <4% between all MOPITT CO
column products (TIR, NIR and JNT) and TROPOMI. TROPOMI CO values were slightly lower
than MOPITT in most regional comparisons.
**3.2 The Total Carbon Column Observing Network (TCCON)**
TCCON is a network of ground-based Fourier Transform Spectrometers that records direct
solar spectra in the NIR spectral region (Wunch et al., 2011; Laughner et al., 2023). TCCON data
has been previously used to evaluate MOPITT products (e.g., Hedelius, et al., 2019). Column-
averaged mixing ratios of chemical species such as $CO_2$, $CH_4$, $N_2O$, and CO are retrieved from
these spectra. We use CO column data from the TCCON GGG2020 data release
(https://tccondata.org/2020; TCCON Team, 2022) to evaluate model results. Data from 18
TCCON sites are used (Buschmann et al., 2022; García et al., 2022; Hase et al., 2022; Iraci et al.,
2022; Kivi et al., 2022; Liu et al., 2022; Morino et al., 2022a, 2022b, 2022c; Notholt et al., 2022;
Pollard et al., 2022; Shiomi et al., 2022; Té et al., 2022; Warneke et al., 2022; Wennberg et al.,
2022a, 2022b; Wunch et al., 2022). We interpolate model results to TCCON data locations and
time and apply TCCON averaging kernels to model results for proper comparisons.
**3.3 NOAA Carbon Cycle Greenhouse Gases (CCGG) sites**
We use the atmospheric CO dry air mole fractions from the NOAA GML Carbon Cycle
Cooperative          Global          Air          Sampling          Network
(https://gml.noaa.gov/aftp/data/trace_gases/co/flask/surface/; Petron et al., 2022). Event data are
used. The reference scale is WMO CO_X2014A. We interpolate model results to CCGG site
locations and time for proper comparisons. Note that on average, each site only has data on ~4
days and ~9 data points in total from July 16th, 2018 to August 14th, 2018.
**3.4 In-service Aircraft for a Global Observing System (IAGOS)**
IAGOS is a European research infrastructure developed for operations on commercial
aircraft to monitor atmospheric composition (Petzold et al., 2015). The IAGOS instrument package
1 measures CO as well as $O_3$, air temperature, and water vapor (https://www.iagos.org/iagos-core-
instruments/package1/). CO is measured by infrared absorption using the gas filter correlation
technique (Precision: ±5%, Accuracy: ±5 ppb). Here we use vertical profiles of CO from IAGOS
for model evaluation. We use CO profiles in North and West Africa, Tropical Asia, East Asia,
Europe, Eastern North America, Western North America, Central and South America, and Middle
East and conduct evaluation in these regions separately. CO profiles used and regions is shown in
Figure S2. Note that IAGOS profiles are divided into regions based on their locations, however
the IAGOS profiles in a region are not representative of the whole region due to coverage (Figure
S2).
**3.5 Western wildfire Experiment for Cloud chemistry, Aerosol absorption and Nitrogen**
**(WE-CAN)**
The WE-CAN field campaign was conducted over the Northwestern U.S. during July–
September 2018 (https://data.eol.ucar.edu/project/WE-CAN). There were 16 research flights of
the NCAR/NSF C-130 research aircraft during the campaign. Our experiment runs start on July
$16^{th}$ and end on August $14^{th}$. Therefore, we compare the model results to measurements from
flights on July-31, August-02, August-03, August-06, August-08, August-09, and August-13. We
use 1-minute averaged CO (Picarro G2401-mc) data. Model results are interpolated to match
locations and time of the observations, and then both interpolated model results and observations
are averaged back to the model spatial resolution (1.25° in longitude and 0.95° in latitude), 6-
hourly bins, and 50 hPa vertical layers. This is because the model spatial and temporal resolution
are much lower than observations and model results cannot reproduce the high variability in the
raw observations.
**4. Diagnostics of the assimilation results**
**4.1 Observation space diagnostics**
**4.1.1 Fractions of observations rejected by the assimilation system**
In all the five experiments, the assimilation improves the agreement between model
forecast and observations of not only the MOPITT products assimilated but also the MOPITT
products that were not assimilated. Assimilating MOPITT CO column product(s) improves model
agreement with MOPITT CO profile product(s) and vice versa. Figure 4 shows time series of the
fraction of observations rejected by the assimilation system (%) when they are too far from the
model ensemble mean. The decreasing fractions with time indicate more observations being
accepted by the model, i.e., and observations and modeled values are getting closer in later time
steps. For a MOPITT product that is not assimilated in an experiment run, it is still used in the
"evaluation mode", where the ensemble is run through the observation operator, but not
assimilated. Therefore, the hypothetical fraction of observations rejected is still calculated for the
MOPITT product for that experiment run, even though these observations are not assimilated. For
the spin-up/control run, there is no significant trend for the fractions of rejected observations
(Figure 4f). For the five experiments, the fractions of rejected observations decrease with time.
Assimilating (Figures 4a-4e) any MOPITT product(s) improves model agreement with all the five
MOPITT CO products regardless if they are column or profile products. When only assimilating
column products (Exp1-CJ; Exp3-CT; and Exp4-CT+CN), the fraction of rejected observations
decreases faster than that when assimilating both profile and column products (Exp5-PT+CN). For
experiments that assimilate profiles (Exp2-PJ and Exp5-PT+CN), the fractions of rejected
observations decrease slower than the other three experiments that only assimilate column
products (Exp1-CJ, Exp3-CT, and Exp4-CT+CN). This is expected because profile assimilation
has relatively small impact than column assimilation overall due to vertical localization.
**4.1.2 Reduced centered random variable (RCRV) and chi-square statistics $\chi^2$**
We use the RCRV as a diagnostic of the ensemble bias (Candille et al., 2007) and has been
previously used to validate assimilation results (e.g., Gaubert et al., 2014). Mean RCRV for P
observations is defined by the ratio between the innovation and its associated error:
$$RCRV = \frac{1}{P}\sum_{i=1}^{P} \frac{y_i^o - \overline{Hx_i^f}}{\sqrt{\sigma_{o,i}^2 + \sigma_{f,i}^2}} \tag{3}$$

Where $y_i^o$ is the value of i-th observation, $\overline{Hx_i^f}$ gives the expected observation from the model, $\sigma_{o,i}^2$
is the observation error variance, and $\sigma_{f,i}^2$ is the ensemble variance. The mean of the RCRV
represents the weighted bias of the forecast, and hence a value close to 0 indicates the ensemble is
representative (i.e., error variances are comparable to the innovations). Figure 5 shows daily
$\overline{RCRV}$. For a given experiment, only $\overline{RCRV}$ of MOPITT product(s) assimilated in the experiment
is shown here. In most cases $\overline{RCRV}$ is close to zero, indicating that the ensemble is representative.
The only exceptions are NIR column product in Exp4-CT+CN and Exp5-PT+CN.
Chi-square statistics ($\chi^2$) is also used to verify an effective assimilation by comparing error
specifications and their balance with actual model-observation mismatch (Ménard and Chang,
2000) and has been previously used to evaluate assimilation results (e.g., Gaubert et al., 2016;
Sekiya et al., 2021). Mean RCRV for P observations is defined as
$$\overline{\chi^2} = \frac{1}{P}\sum_{i=1}^{P} \frac{(y_i^o - H\underline{x}_i^f)^2}{\sigma_{o,i}^2 + \sigma_{f,i}^2} \tag{4}$$

A value lower than 1 indicates an overfitting of the observations while a value higher than 1
suggests an underestimation of the actual model and observation mismatch. Daily $\overline{\chi^2}$ are also
shown in Figure 5. The $\overline{\chi^2}$ values are all higher than 1 indicating an underestimation of the actual
model and observation mismatch. However, $\overline{\chi^2}$ decreases with time and gradually approaches
towards 1, indicating the degree of such underestimation decreases with time.
**4.2 Model space diagnostics**
We analyze the impacts of assimilating MOPITT CO products by comparing the
experiment runs with control/spin-up run, which effectively isolate the signal resulting from the
CO assimilation. Figure 6 show the spatial distribution of CO difference caused by assimilation
(CO from forecast of experiment minus CO from the control/spin-up run) for the 5 experiments
(15-day average). At the surface, the spatial distributions of CO difference are similar among the
5 experiments. In line with Gaubert et al. (2023), the 5 experiments show overall higher CO in the
Northern Hemisphere and lower CO in the tropics and India compared to the control/spin-up run.
Exp2-PJ and Exp5-PT+CN reduce CO in California which is not the case for other experiments.
Exp2-PJ and Exp5-PT+CN are the only two experiments that involves profile product assimilation.
In addition, profile JNT is retrieved with profile TIR and column NIR therefore Exp2-PJ is
expected to assimilate similar information as Exp5-PT+CN. In addition, when comparing Exp1-
CJ and Exp1-PJ, column assimilation has a larger downwind impact (e.g., the ocean between
Africa and South America). At 500 hPa, the 5 experiments still show overall higher CO in the
Northern Hemisphere compared to the control/spin-up run. However, the Exp2-PJ and (5) that
involve profile assimilation have lower CO values than the other 3 experiments, especially in the
high latitudes. At 200 hPa, the spatial distribution of the CO difference caused by assimilation is
smallest in Exp2-PJ, followed by Exp5-PT+CN. On the contrary, for the other three experiments
which do not involve profile assimilations, the spatial distribution of the CO difference caused by
assimilation is relatively large, i.e., assimilating MOPITT profile product(s) only slightly changes
CO values at 200 hPa whereas assimilating MOPITT column product(s) changes CO values at 200
hPa dramatically. This is expected as vertical distribution is often an advantage of profile DA that
column DA cannot represent.
Assimilating profile products have different vertical impacts from assimilating column
products (Figure 7). Overall, the two experiments that involve profile assimilation (Exp2-PJ and
Exp5-PT+CN) seem to be close to each other, while the other three experiments that only involve
column assimilation (Exp1-CJ, Exp3-CT, and Exp4-CT+CN) also exhibit similarities among
themselves. Globally speaking, experiments that assimilate only column product(s) have a larger
impact at and near the surface compared to experiments that assimilate only profile product(s)
(Figures 7a and 7b). This is reasonable because profile assimilation is more localized vertically.
Regional speaking, the impacts of the five experiments vary across continents.
The difference caused by assimilating profile products is in general smaller than the
difference caused by assimilating column products. The exceptions are Africa and South America
where the two experiments that assimilate profiles have lower CO than the three experiments that
only assimilate columns between 900 hPa and 600 hPa. CO over the two regions is dominated by
fire emissions during the experiment period. It is known that FINN overestimates fire emissions
in the tropics (Wiedinmyer et al., 2023; Gaubert et al., 2023) of CO which were transported to
upper levels through fire plume rise and tropical convection. This overestimation between 900 hPa
and 600 hPa is corrected by assimilating MOPITT CO products, especially profile products that
captured CO plumes between 900 hPa and 600 hPa. Exp2-PJ and Exp5-PT+CN have some
relatively small differences over some regions even though profile JNT is retrieved with profile
TIR and column NIR. For example, over North America, Exp2-PJ has lower CO values than Exp5-
PT+CN. Exp1-CJ and Exp4-CT+CN are in general similar with some exceptions. For example,
over Africa between 900 hPa and 600 hPa, CO profile from Exp1-CJ is closer to Exp3-CT rather
than Exp4-CT+CN.
**5 Comparisons with independent observations**
**5.1 TROPOMI**
To evaluate the results, we compare the CO from DA forecasts with independent
observations. Comparisons with TROPOMI CO column retrievals are shown in Figure 8. The
control run underestimates background CO in the Northern Hemisphere while overestimates CO
near fire source regions in the tropics and Southern Hemisphere. Compared to the control run, all
five of the experiments show improved agreement with TROPOMI CO by increasing background
CO in the Northern Hemisphere and reducing CO near fire source regions in the tropics and
Southern Hemisphere. The spatial distributions of the mean biases from the three experiments with
only column assimilation are close while those from the two experiments with profile assimilation
are close. The two experiments with profile assimilations have smaller improvement for
background CO in the Northern Hemisphere. This is reasonable because profile assimilation has
relatively small impact than column assimilation due to tight vertical localization. However, near
the fire source regions, the two experiments with profile assimilations have lower biases than the
three experiments with only column assimilation. This is the case not only in Africa, South
America and tropical Asia (Figure 8), but also in California (fire region) and Nevada (downwind
of the fire region), USA during the study period which is the fire season in the region (Figure S5).
This indicates profile assimilation can out-perform column assimilations in circumstances with
fire impacts, which is likely due to transport errors and fire plume rise that requires vertical
information to resolve plume locations.
**5.2 TCCON**
Overall, the control run tends to underestimate CO and the 5 experiments all agree better
with TCCON observations compared to the control run but still underestimates CO in general
(Figure 9). Column assimilations (Exp1-CJ, Exp3-CT, and Exp4-CT+CN) significantly
overestimate CO at pasadena01 and edwards01 sites in California, USA during 26 July 2018 to 04
August 2018, likely due to fire impacts. The significant overestimation is not seen in the two
experiments with profile assimilations (Exp2-PJ and Exp5-PT+CN). This is consistent with the
comparison results with TROPOMI and implies the profile assimilation can out-perform column
assimilations in fire-impacted regions. The model-observation discrepancies overall decrease with
time. A time series of TCCON and modeled CO columns is shown in Figure S6.
**5.3 CCGG sites**
All experiments show improved agreement with surface in-situ CO observations from
CCGG sites compared to the control run (Figure 10), as shown by with higher correlations (0.6-
0.65 versus 0.56) and lower model biases (0.7-4.91 ppb versus 8.6 ppb). As for RMSE, however,
the experiments do not reduce RMSE compared to the control run (34-50 ppb versus 36 ppb).
Exp1-CJ has the lowest mean bias (5.7 ppb) while Exp5-PT+CN have the highest correlation
(0.79).
Spatial distributions of model bias in CO (ppb) against CO observations from CCGG sites
are shown in Figures S7-S10. The UTA CCGG site is close to the two TCCON sites in California,
USA (pasadena01 and edwards01). All the five experiments significantly underestimate CO at the
UTA surface site during 26 July 2018 to 4 August 2018, whereas the five experiments overestimate
CO compared to the two TCCON sites (Figure 9). This inconsistency is likely due to (1) UTA
CCGG site measures CO at the surface while the TCCON sites measure column total CO; (2) there
are only two data points during that period at the UTA site and are not comparable to the sampling
of the two TCCON sites.
**5.4 IAGOS**
Globally, all five experiments agree better with IAGOS CO profiles compared to the
control run (Figure 11a). At the 900-1000 hPa layer, Exp2-PJ has the lowest bias, followed by
Exp4-CT+CN. At layers above 800 hPa, the three experiments with only column assimilation have
lower bias. CO bias of Exp1-CJ and Exp4-CT+CN are very similar using that of Exp3-CT as a
reference. This is expected as Column JNT product contains similar information as column TIR
product and column NIR products together. Above 200 hPa, all five experiments overall agree
better with IAGOS CO compared to the control run. However, experiments involving profile
assimilation do not show obvious differences compared to experiments only involving column
assimilation above 200 hPa. Over most regions, the five experiments show improved agreement
with IAGOS data except for Tropical Asia and Central and South America where the five
experiments have similar or larger biases (Figure 11). Over North and West Africa, the control run
has positive bias whereas the five experiments have negative biases below 500 hPa, indicating the
system might over-adjust in the region. The comparisons with IAGOS show that the experiments
overall perform better in the Northern Hemisphere than in the tropics.
**5.5 WE-CAN**
The experiments do not show improvement from the control run when compared to
airborne measurements from WE-CAN. This is expected because the airborne measurements
during WE-CAN aimed to sample fire plumes and include extremely high CO concentrations
which are challenging for a 1-degree global model to capture, not to mention the output is 6-hourly.
The experiments only do show lower model bias than the control run (-24 to -48 ppb versus -52
ppb), however the difference between Exp2-PJ and Exp5-PT+CN from the control run is small.
The correlation and RMSE of the experiments are not improved. The subtle improvement in the
mean bias is likely driven by large-scale adjustment rather than improvement in resolving flight-
scale features.
**6. Emissions**
**6.1 Emission updates**
Assimilating profile products (Exp2-PJ and Exp5-PT+CN) tends to have a larger change
to the emissions compared to only assimilating column products (Exp1-CJ, Exp3-CT, and Exp4-
CT+CN). As shown previously, profile assimilation can out-perform column assimilations near
the surface due to vertical localization. Different CO concentrations at and near the surface resulted
in different emission updates between profile assimilation and column assimilation. The 5
experiments overall increase anthropogenic CO emissions while reduce fire CO emissions (Figure
13). For anthropogenic emissions, the two experiments that assimilate CO profiles (Exp2-PJ and
Exp5-PT+CN) significantly increase anthropogenic CO emissions from ~500 Tg/year to ~700
Tg/year globally in August, which is not the case for the other experiments. Anthropogenic
emissions in India are reduced by the experiments while in East Asia are increased (Figure 14).
Fire emissions are reduced by the 5 experiments in Africa and South America and the reduction is
the largest for the two experiments that assimilate CO profiles (Figures 13 and 14). This is
consistent with the conclusion in Wiedinmyer et al. (2023), which found fire emissions in
FINNv2.4 over Africa are too high, and consequently were reduced in FINNv2.5. The experiments
overall increase fire emissions in North America, indicating that FINNv2.4 underestimates fire
emissions in the region during the assimilation period. Fire and anthropogenic emissions can have
different injection heights and impact different vertical levels. This is especially the case for
regions with strong convection (e.g., central Africa).
**6.2 CAM-chem simulations with updated emissions**
We compared the CAM-chem simulations with updated emissions and original emissions
to CO observations from TROPOMI, TCCON, CCGG site, IAGOS, and WE-CAN (Figures S11-
S18). The five simulations with updated emissions overall show better agreement with
observations compared to the control run with original emissions. Simulations using emissions
from profile assimilation experiments (Simulations (S2) and (S5)) in general perform better than
column assimilation especially near the surface (S17) and at fire source regions (Figures S11, S12,
and S14). This is consistent with the evaluation of DA experiments. This indicates assimilating
satellite profiles can perform better near the surface and have a larger impact on emissions
compared to only assimilating column products.
**7. Discussions**
**7.1 Assimilating multispectral product versus TIR-only product**

The comparisons between Exp1-CJ and Exp3-CT demonstrate the impacts of assimilating
satellite multispectral/joint products versus TIR-only products. Overall, when comparing to
independent CO column observations, assimilating joint products do not show clear improvement
from assimilating TIR-only products (Figures 8 and 9). However, when comparing to independent
CO profile observations or surface CO observations, assimilating joint products leads to better
model-observation agreement at and near the surface (Figures 10 and 11). This is reasonable as
the joint MOPITT product has enhanced sensitivity to near-surface CO (Worden et al., 2010).
**7.2 Assimilating profile product versus column product**
The comparisons between Exp1-CJ and Exp2-PJ demonstrate the impacts of assimilating
satellite multispectral/joint products versus TIR-only products. The fractions of rejected
observations for Exp3-CT decrease slower than Exp1-CJ due to vertical localization when
assimilating profile products. For the same reason, assimilating column products has a larger
impact on the analysis compared to assimilating profile products. Therefore, Exp2-PJ with profile
assimilation has smaller improvement for background and large-scale CO in the northern
hemisphere (Figure 8) compared to Exp1-CJ with column assimilation. However, assimilating
profile products can have different vertical impacts from assimilating column products (figure 7).
Profile assimilation can out-perform column assimilations in fire-impacted regions and near the
surface (Figure 11).
Assimilating profile products tends to have a larger change to the emissions compared to
only assimilating column products. Simulations using emissions from profile assimilation
experiments in general perform better than column assimilation especially near the surface and at
fire source regions.
**7.3 Assimilating multispectral product versus assimilating TIR and NIR separately**
For multispectral/joint products, we also compare the impacts of assimilating the joint
product directly versus assimilating the single spectral products separately. MOPITT column JNT
products are retrieved from MOPITT column TIR and column NIR products, while MOPITT
profile JNT products are retrieved from MOPITT profile TIR and NIR products. Therefore, we
compare Exp1-CJ to Exp4-CT+CN, Exp2-PJ to Exp5-PT+CN for demonstration. In general,
assimilating multispectral/joint products result in similar or slight better agreement with
observations compared to assimilating the single-spectral products separately. This is the case for
both assimilating profile products (Exp2-PJ versus Exp5-PT+CN) and column products (Exp1-CJ
versus Exp4-CT+CN). In addition, assimilating multispectral/joint products is more
computationally efficient than assimilating single spectral products separately. These two reasons
point to the benefit of developing multispectral/joint products for CO as well as other species such
as $O_3$ and $CH_4$ and assimilating them in DA systems.
**7.4 Limitation**
Here we only conduct experiments for 15 days as the number of experiments and
computational cost prohibit longer simulations. A previous study performed longer simulations for
one experiment that assimilated the MOPITT profile product for a whole year (Gaubert et al.,
2016) and found that there is no significant seasonal change in the performance of the CAM-
chem+DART. If observations of roughly the same quality/quantity are available in other years, the
performance of the DA might be expected to be similar. However, more research is needed to fully
understand the impact of (1) assimilating multispectral/joint products versus single-spectral

products, (2) the comparison of satellite profiles and satellite columns DA, and (3) assimilating multispectral or each product separately. This study provides guidance for future work on the assimilation of multi-spectral satellite retrievals of atmospheric composition using MOPITT as a demonstration. However, whether the conclusions based on MOPITT CO are applicable to other species (e.g., CH4 and O3) needs further study. Nevertheless, the results and conclusions presented in this study are valid and shed light on the impacts of assimilating different satellite products of the same atmospheric composition.

The CAM-chem+DART experiments in this study overall show improvement in background and large-scale CO distributions compared to the control/spin-up run, as shown by the comparisons with global observations such as TROPOMI and TCCON. However, CAM-chem+DART improvement on small-scale features is challenging due to limitation in model resolution, as shown by the comparisons with airborne measurements during WE-CAN. A higher resolution DA system is needed to resolve these features. We are currently developing the capability of DA using MUSICA+DART which will address this issue (Pfister et al., 2020). MUSICA has already been shown to better resolve fires at higher resolution while still addressing global-scale impacts (Tang et al., 2022, 2023).

## 8. Conclusions

We conduct 6 CAM-chem+DART assimilation runs for 15 days (July 31st, 2018 to August 14$^{th}$, 2018) to understand the impact of (1) assimilating multispectral products versus single-spectral products, (2) assimilating satellite profile products versus column products, and (3) assimilating multispectral products versus assimilating individual products separately. The DA runs include 1 control run that only assimilates meteorological variables and 5 experiment runs that assimilate meteorological variables and different MOPITT product(s), namely Exp1-CJ; Exp2-PJ; Exp3-CT; Exp4-CT+CN; and Exp5-PT+CN. We then compare the results with independent CO observations from satellite, ground-based remote sensing, surface and aircraft observations (TROPOMI, TCCON, CCGG sites, IAGOS, and WE-CAN). Fire and anthropogenic emissions of CO are also optimized in the DA experiments. We conduct 5 CAM-chem runs with the 5 sets of optimized emissions to understand the impacts of assimilating different MOPITT products. We also conduct 1 additional CAM-chem runs with original emissions for reference. The main findings are as follows:

(1) Assimilating MOPITT profile products improves model agreement with MOPITT column products and vice versa.

(2) All five DA experiments show improved agreement with CO observations from TROPOMI, TCCON, CCGG sites, and IAGOS compared to the control/spin-up run. Assimilating MOPITT joint column product leads to better model-observation agreement at and near the surface than assimilating MOPITT TIR-only column product.

(3) Assimilating profile products tends to have a larger change to the emissions compared to only assimilating column products. The five experiments overall increase anthropogenic CO emissions while reducing fire CO emissions. The five CAM-chem simulations with updated emissions overall show better agreement with observations compared to the control run with original emissions. Simulations using emissions from profile assimilation experiments in general perform better than column assimilation especially near the surface and at fire source regions.

(4) Assimilating column products has larger impacts and improvement for background and large-scale CO compared to assimilating profile products due to vertical localization in profile

assimilation. However, profile assimilation can out-perform column assimilations in fire-impacted regions and near the surface.

(5) Assimilating multispectral/joint products result in similar or slightly better agreement with observations compared to assimilating the single-spectral products separately. Assimilating multispectral/joint products is also more computationally efficient than assimilating single spectral products separately. Therefore, it is advantageous to develop multispectral/joint products for CO as well as other species (e.g., $O_3$ and $CH_4$) and assimilating them in DA systems.

**Competing interests**

At least one of the (co-)authors is a member of the editorial board of Atmospheric Measurement Techniques.

**Acknowledgement**

This project is partially supported by NOAA Atmospheric Chemistry, Carbon Cycle and Climate (AC4) Program (Award Number: NA22OAR4310204). This material is based upon work supported by the National Center for Atmospheric Research, which is a major facility sponsored by the National Science Foundation under Cooperative Agreement No. 1852977. We would like to acknowledge high-performance computing support from Cheyenne (doi:10.5065/D6RX99HX) provided by NSF NCAR's Computational and Information Systems Laboratory, sponsored by the National Science Foundation. We thank TROPOMI, TCCON, NOAA CCGG, IAGOS, and WE-CAN team for observational data. The TCCON data were obtained from the TCCON Data Archive hosted by CaltechDATA at https://tccondata.org.

**Author contribution**

Conceptualization, HMW; Investigation, WT and BG; Methodology, BG, WT, HMW, and LKE; Formal analysis, WT and BG; Data curation, DZ, DM, KR, and JLA; Validation, WT; Visualization, WT; Supervision, HMW; Writing – original draft preparation, WT, BG, and HMW; Writing – review & editing, LKE, DPE, AFA, DZ, DM, KR, and JLA.

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

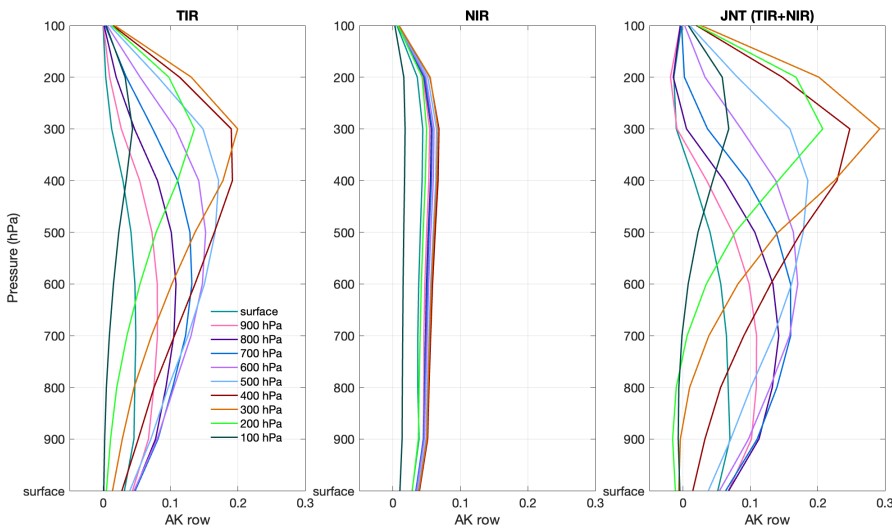

**Figure 1**. Averaging kernel (AK) rows for MOPITT retrieval types TIR only, NIR only, and
multispectral TIR+NIR. Global average of AKs during July and August 2018 are shown.

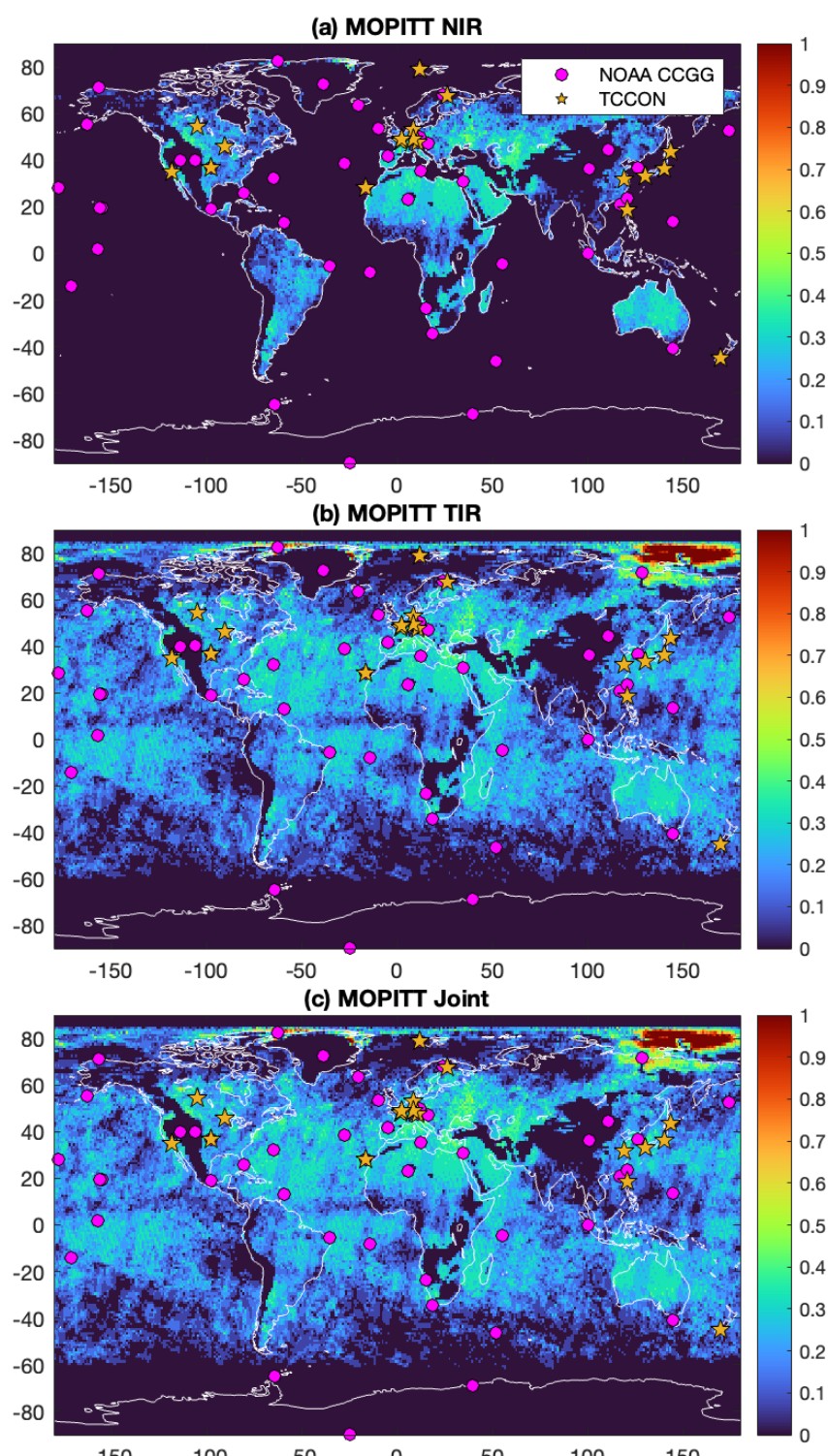

**Figure 2**. Daily number of super-observations per day and per grid from MOPITT (a) TIR, (b)
NIR, and (c) JNT products during July 16th 2018 to August 14th 2018. Total Carbon Column
Observing Network (TCCON) sites are marked by yellow stars and NOAA Carbon Cycle
Greenhouse Gases (CCGG) sites are marked by pink circles.


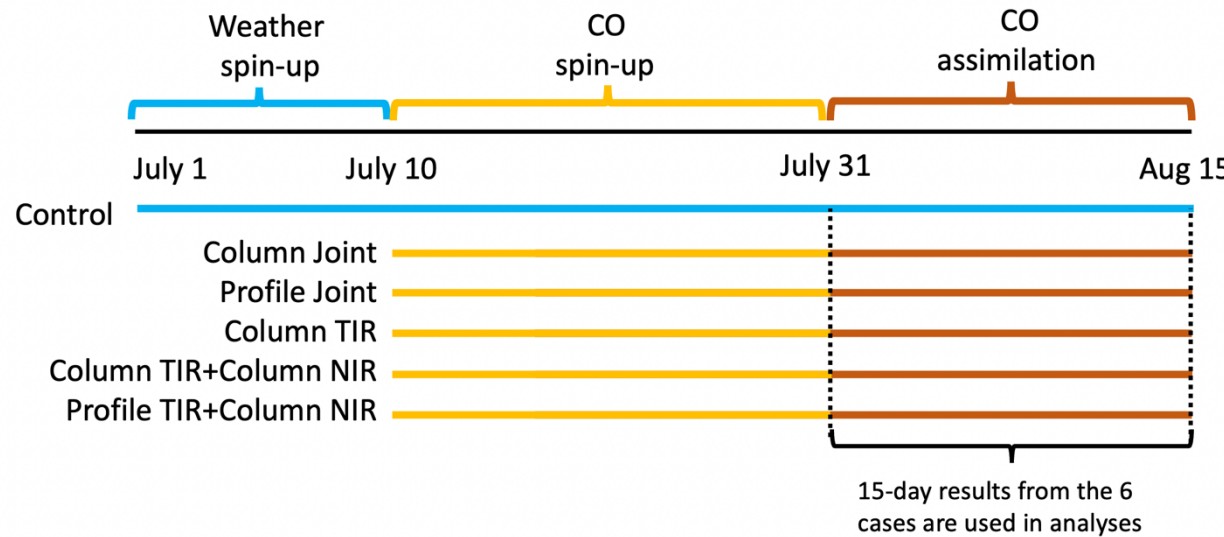

**Figure 3.** Setup of the CAM-chem/DART data assimilation experiments.

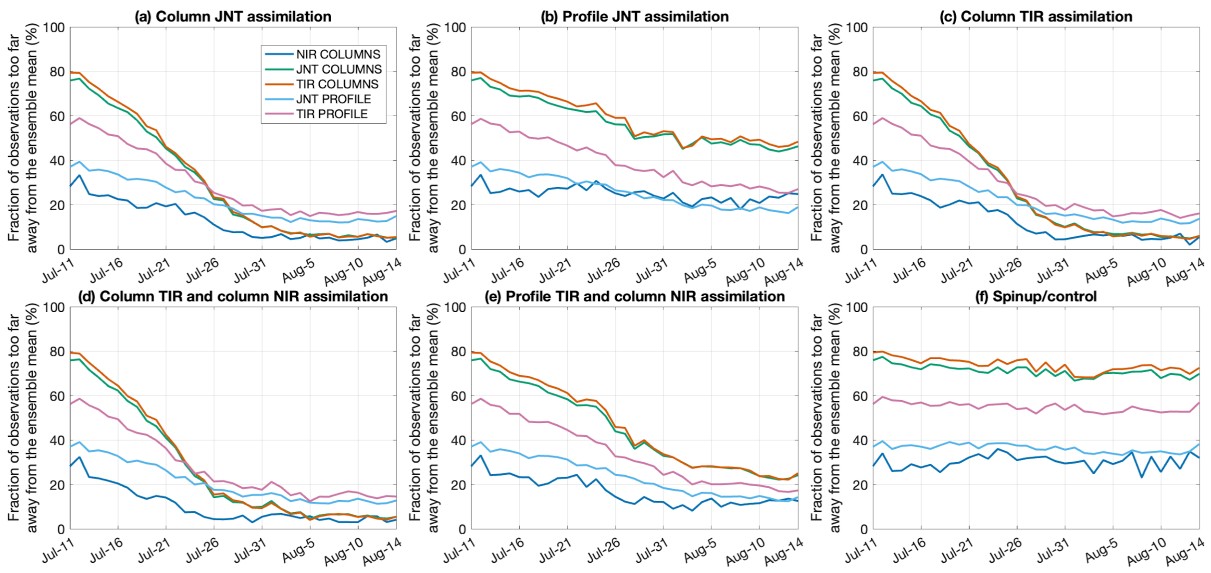

**Figure 4**. Time series of the fractions of observations rejected by the assimilation system (%) due
to that they are too far from the ensemble mean.

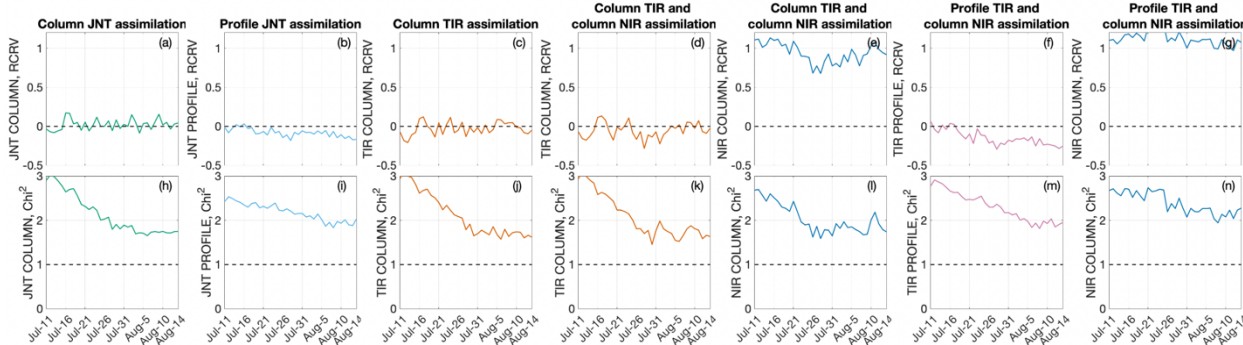

**Figure 5**. timeseries of (a-g) daily mean of Reduced Centered Random Variable (RCRV) and (h-
n) daily mean of Chi-square. For each experiment, only RCRV and Chi-square of the MOPITT
product that were assimilated are shown.

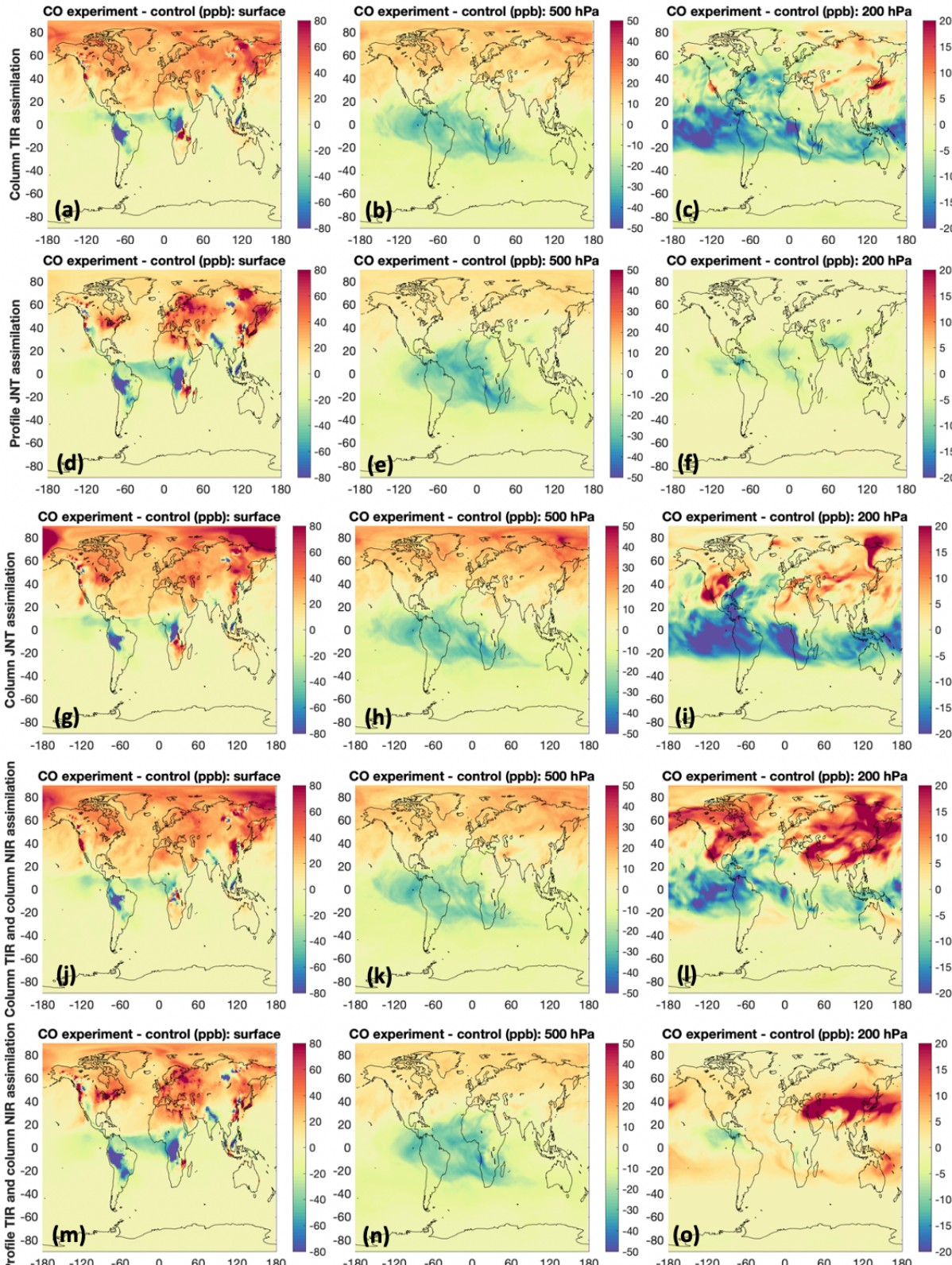


**Figure 6**. 15-day (July 31 - August 14, 2018) average of the difference in CO (forecast of
experiment minus control run) for the 5 experiments at the model surface, 500 hPa, and 200 hPa.
Note that the color scales for model surface, 500 hPa, and 200 hPa are different.

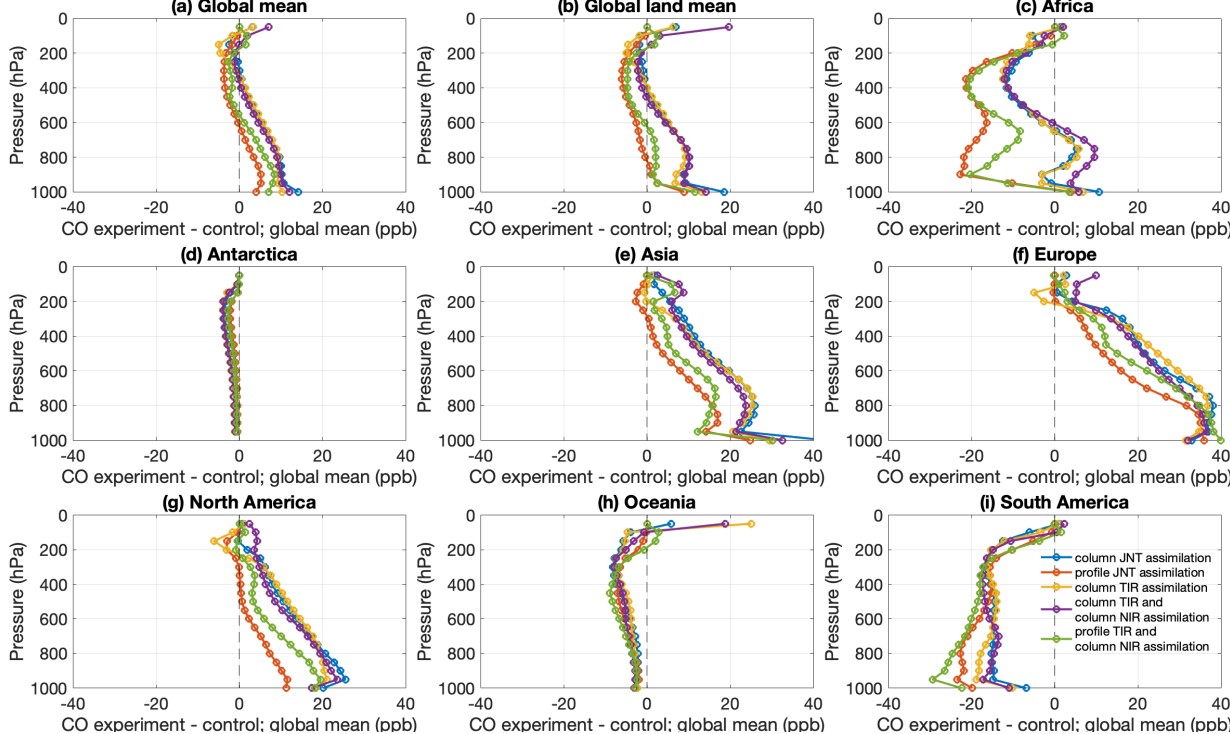

**Figure 7**. Vertical profile of the 15-day (July 31 - August 14, 2018) average difference in CO
(forecast of experiment minus control run) over different regions.

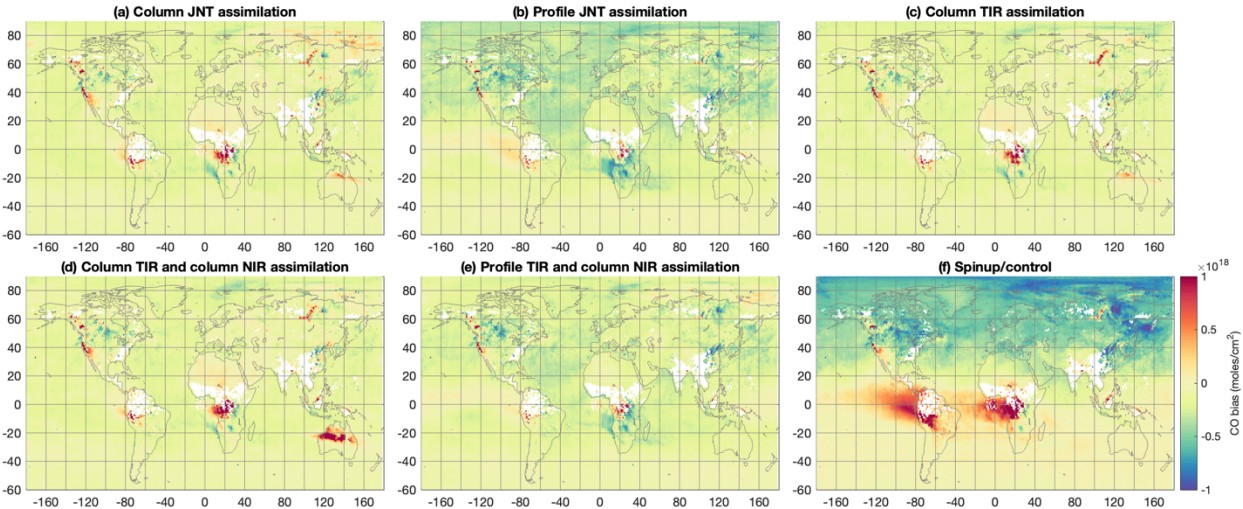

**Figure 8**. 15-day (July 31 - August 14, 2018) mean biases (ppb) of modeled CO against CO
columns from the TROPOspheric Monitoring Instrument (TROPOMI) for the 5 experiments and
the control run. TROPOMI averaging kernels are applied to model CO for the comparisons.


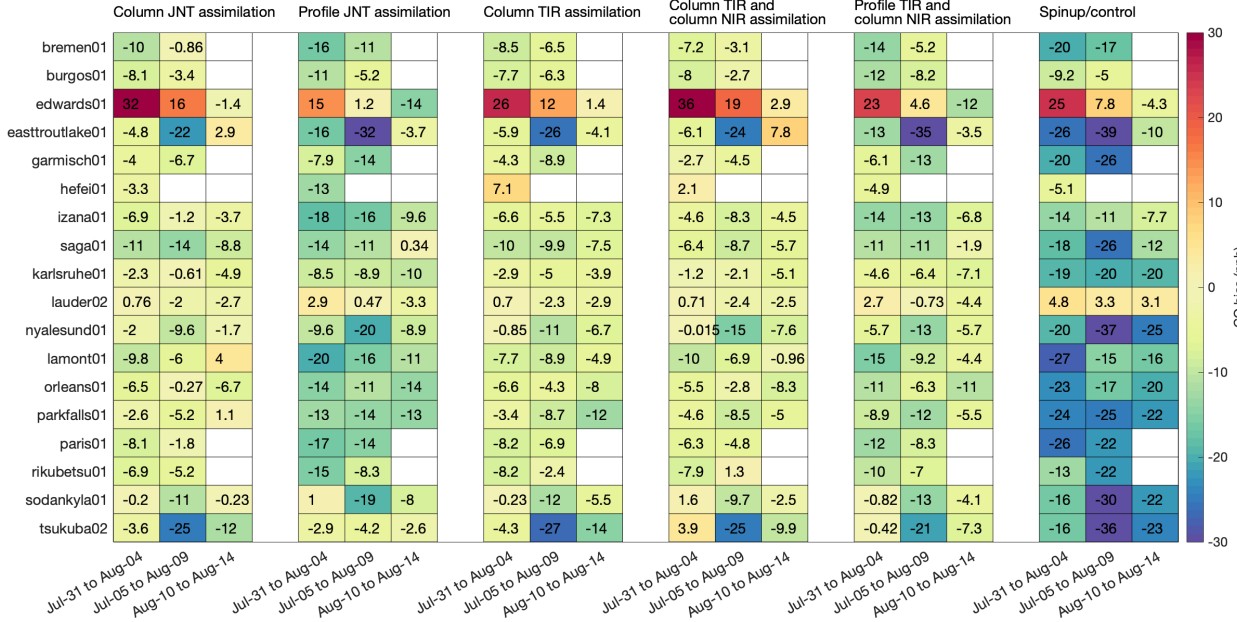

**Figure 9**. Mean biases (ppb) of modeled CO against CO columns from the Total Carbon Column
Observing Network (TCCON) for the 5 experiment and the control run. TCCON averaging kernels
are applied to model CO for the comparisons. Spatial locations of TCCON sites can be found in
Figure 3 and Figure S1. A time series of TCCON and modeled CO can be found in Figure S4.

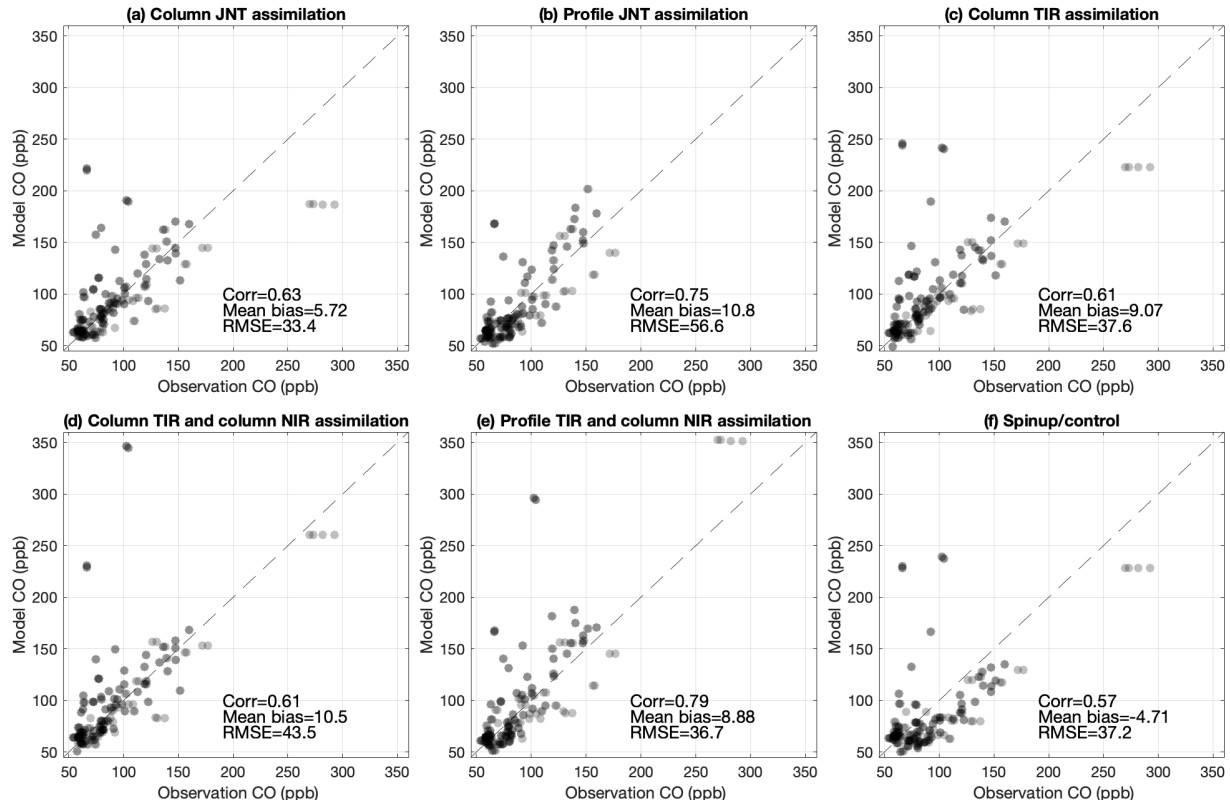

**Figure 10**. Comparisons of modeled CO (ppb) and CO observations (ppb) from the NOAA Carbon Cycle Greenhouse Gases (CCGG) sites during July 31st, 2018 to August 14th, 2018 for the 5 experiments and the control run. Spatial locations of CCGG sites can be found in Figure 3 and Figure S1. A spatial distribution of model bias in CO against CO observations from CCGG sites can be found in Figure S5.

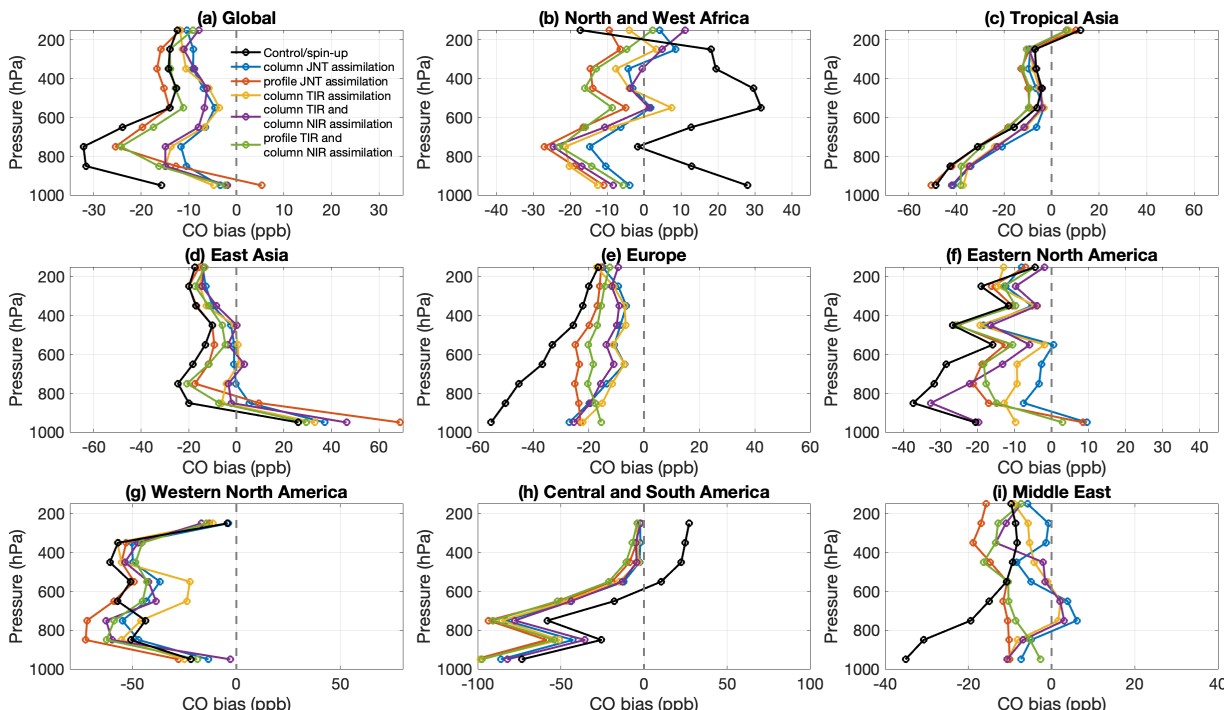

**Figure 11**. Mean biases (ppb) of modeled CO against CO profiles from the In-service Aircraft for
a Global Observing System (IAGOS) measurements for the 5 experiments (colored lines) and the
control run (black line) at different vertical levels. Locations of IAGOS CO profiles can be found
in Figure S2.

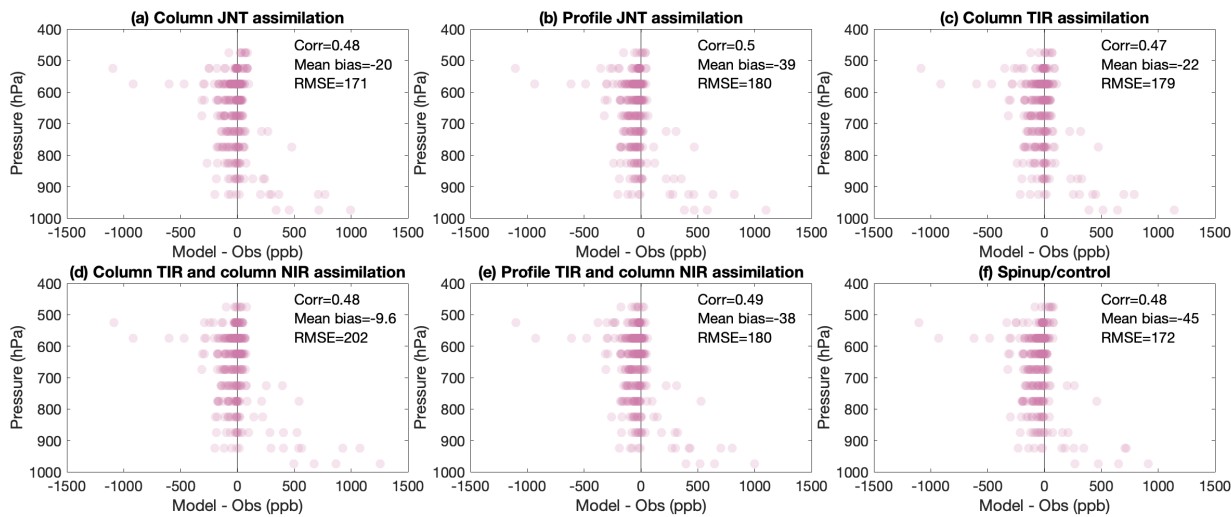

**Figure 12**. Mean biases (ppb) of modeled CO against airborne CO observations from the Western
wildfire Experiment for Cloud chemistry, Aerosol absorption and Nitrogen (WE-CAN) field
campaign for the 5 experiments and the control run at different vertical levels.

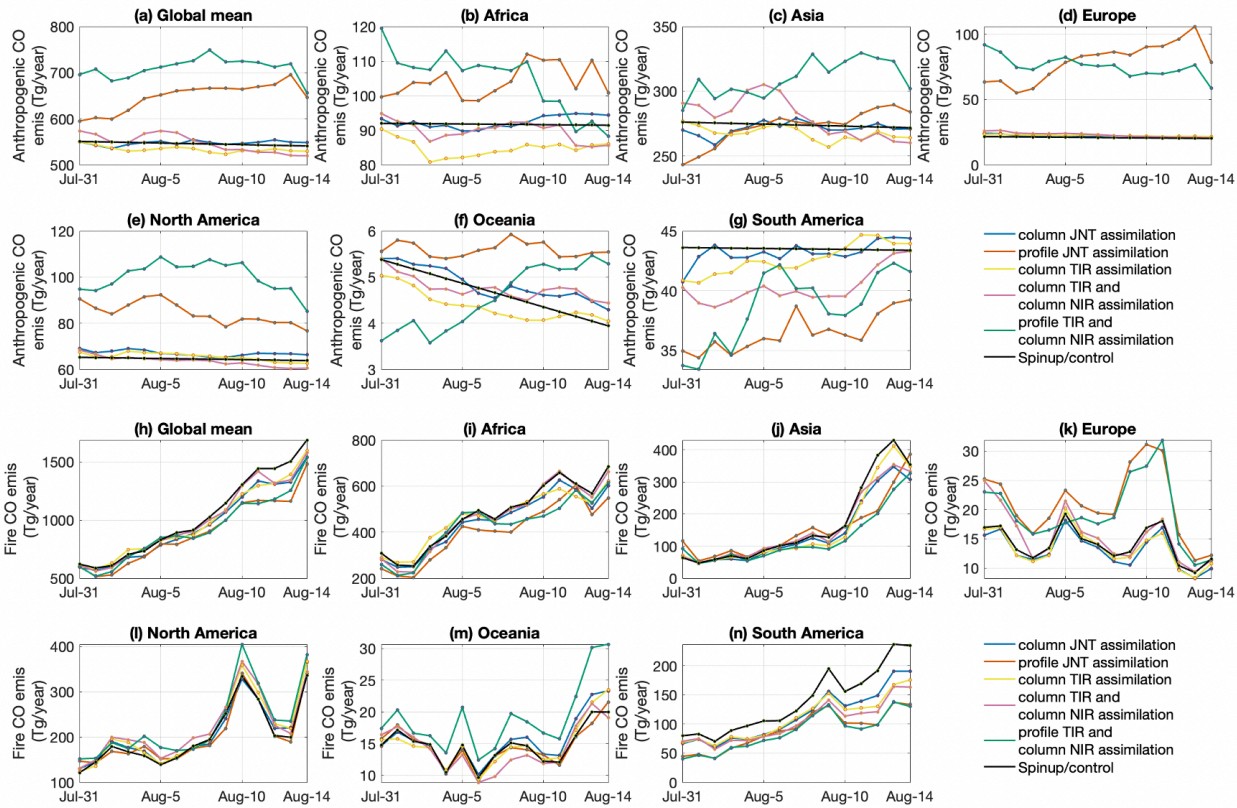

**Figure 13**. Updated (a-g) CAMS anthropogenic CO emissions and (h-n) FINNv2.4 fire CO
emissions as a result of assimilating different MOPITT products. The emissions from the
Spinup/control run are the unchanged original emissions of CAMS and FINNv2.4.

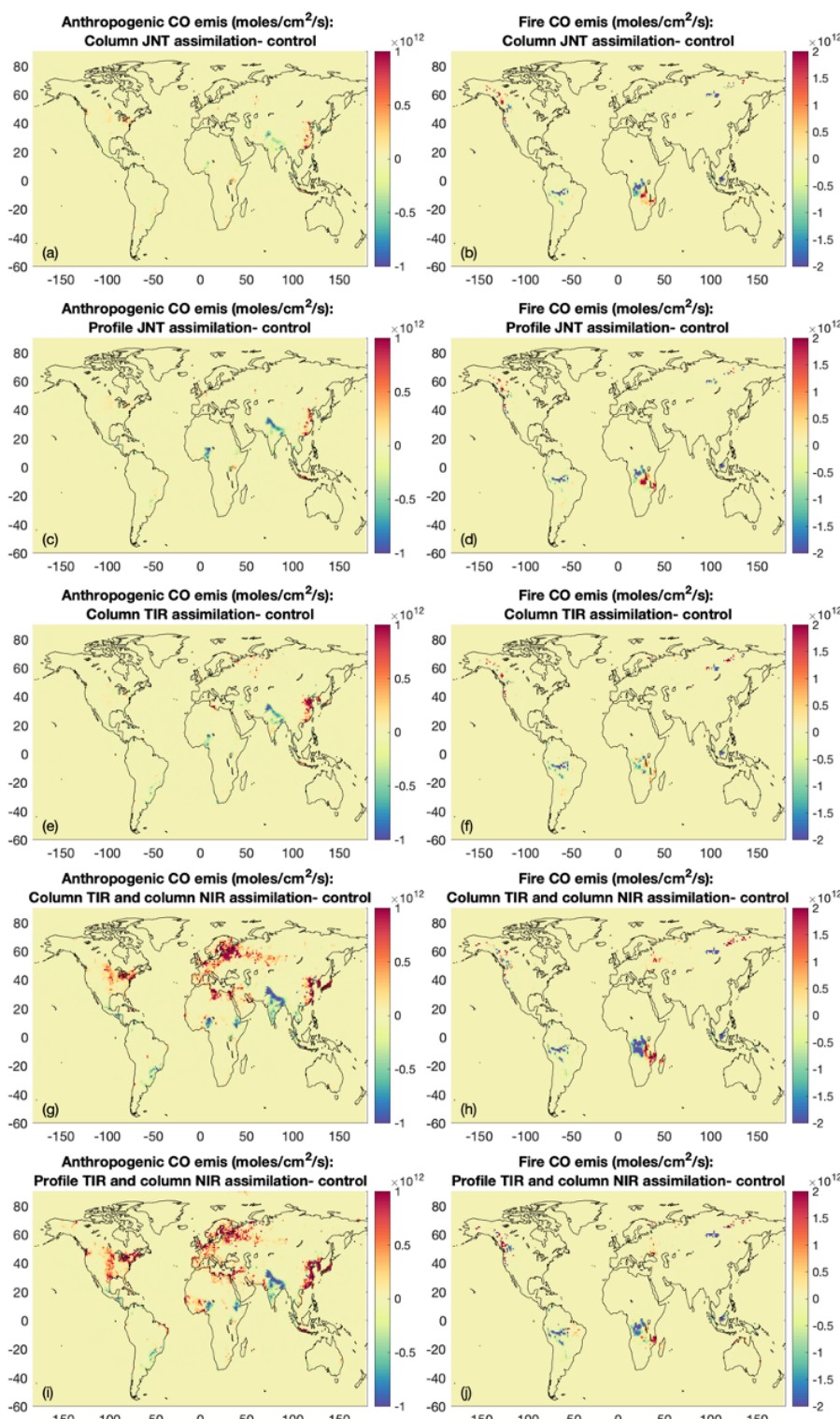

**Figure 14**. Updates on the (a) CAMS anthropogenic CO emissions and (b) FINNv2.4 fire CO
emissions as a result of assimilating MOPITT Column JNT product. Updates is calculated as CO
from the experiment minus CO from the control run. (c-j) are similar to (a-b) but for other
experiments.