# Peer review of "Advantages of assimilating multi-spectral satellite retrievals of atmospheric composition: A"

_Atmospheric Measurement Techniques, 2023_

## Author Comment (AC1)

General comments

1. The study conducted by Tang et al. evaluated the impact of assimilating multispectral/joint versus TIR-only retrieval, column versus profile retrieval, and joint products versus single-spectral products separately, through a 15-day inversion in CAM-chem+DART system, based on MOPITT CO database. They evaluated the assimilation performance against both the assimilated data and independent data (TROPOMI, TCCON, NOAA CCGG, IAGOS, and WE-CAN). Overall, this study is convincing and of great interest to the community studying CO or satellite retrieval, within the scope of AMT.

**Response:** Thank you for spending time reviewing our work.

2. Readers would be benefit from understanding if such comparison remains consistent across other seasons and years. In section 7.4, the authors acknowledge that these results might not be representative for other seasons, and performing a long-term assimilation has high computational cost. Please consider a direct comparison of measurements, rather than their assimilation performance, to evaluate the potential of robustness in discrepancies and similarities across datasets over varying timeframes.

**Response:** This study focused on using assimilation to compare with other observations. We refer to previous direct comparisons of measurements, such as MOPITT and TCCON in Hedelius et al. (2019), and we have added a reference for MOPITT and TROPOMI comparisons, that we neglected previously, to Martínez-Alonso et al., (2020), along with the text:

"TROPOMI CO data were compared to MOPITT CO in Martínez-Alonso et al., (2020). TROPOMI and MOPITT data show good agreement in terms of temporal and spatial patterns with global average biases <4% between all MOPITT CO column products (TIR, NIR and JNT) and TROPOMI. TROPOMI CO values were slightly lower than MOPITT in most regional comparisons."

Note that direct measurement comparisons of remotely sensed satellite observations with IAGOS in situ data are problematic since IAGOS data are at discrete pressures along a flight track without the vertical profile information needed for smoothing with a satellite instrument operator. However, comparisons at the specific IAGOS pressure levels CO distributions after data assimilation are straightforward.

Unfortunately, the number of experiments prohibits longer simulations, and as the reviewer noticed, we discussed this limitation in the text. Nevertheless, a previous study from our group performed longer simulations for one experiment that assimilated the MOPITT profile product for a whole year (Gaubert et al., 2016) and found that there is no significant seasonal change in the

performance of the CAM-chem+DART. We have added this statement to Section 7.4. We also added the following statement to Section 7.4:

"If observations of roughly the same quality/quantity are available in other years, the performance of the DA might be expected to be similar."

Below is Figure 3 of Gaubert et al. (2016).

[Figure]

**Figure 3.** Time series of global assimilation diagnostics in observation space. For every assimilation step, the observations (and statistics) are averaged across the globe (horizontal) and over vertical sections: (left column) lower troposphere (from surface to 700 hPa), (middle column) middle troposphere (from 700 to 400 hPa), and (right column) upper troposphere (lower pressure than 400 hPa). (first row) The CO concentrations in ppb for MOPITT in black, 6-hourly forecast from the Control Run in blue and from MOPITT Reanalysis run in red. (second row) The innovations for both runs. (third row) The chi-square statistics of the forecasts from the MOPITT Reanalysis run. (fourth row) The number of observations (number of observation per 6 h) successfully assimilated in CAM-Chem and for which statistics are calculated.

3. Please clarify the treatment of observation error estimates. Did the inversions include the information of observation uncertainty? Can the observation uncertainty provided by MOPITT associated with measurements, effectively highlight the useful information from each dataset?

**Response:** The assimilations in this study include the information of observation uncertainty. The observation uncertainty provided by MOPITT is associated with measurements, and can effectively highlight the useful information from each dataset. Data assimilation requires observation errors associated with the quantity assimilated. MOPITT provides 3 types of uncertainty estimates: total error, measurement error, and smoothing error. Total error includes both measurement error and smoothing error. Since our observation operators include the smoothing by the MOPITT averaging kernels and the prior profiles, we only use the measurement error rather than total error provided by MOPITT for both column and profile products as smoothing error is already addressed by observation operators in the system. Specifically, for MOPITT profile products, measurement error is provided by the variable "MeasurementErrorCovarianceMatrix" while for MOPITT column products, measurement error is provided by the variable second column of the "RetrievedCOTotalColumnDiagnosticsDay". In addition, within the Data Assimilation Research Testbed, we use adaptive covariance inflation to adjust the total error (model and observations) using the given observation error as reference. Satellite observations are inversions themselves and are quite complex, data assimilation provides an accurate way to assess this information. We have added this information to Section 2.1:

"Data assimilation requires observation errors associated with the quantity assimilated. MOPITT provides 3 types of uncertainties/errors: total error, measurement error, and smoothing error in the products. Total error includes both measurement error and smoothing error. Since our observation operators include the smoothing by the MOPITT averaging kernels and the prior profiles, we only use the measurement error rather than total error provided by MOPITT for both column and profile products as smoothing error is already addressed by observation operators in the system. Specifically, for MOPITT profile products, measurement error is provided by the variable "MeasurementErrorCovarianceMatrix" while for MOPITT column products, measurement error is provided by the variable second column of the "RetrievedCOTotalColumnDiagnosticsDay". In addition, within DART, we use adaptive covariance inflation to adjust the total error (model and observations) using the given observation error as reference."

Specific comments

4. Figure 3: Please explain the criteria used to reject observations too far from the ensemble mean. Providing details on the threshold or methodology employed for this rejection would enhance the transparency of the assimilation process.

**Response:** We have included the following information in Section 2.4:

"Quality Checks are common in data assimilation as the algorithms are employed operationally for near real time forecasting. We use the standard option in DART to do such quality checks. The absolute value of the difference between the observed value and the prior ensemble mean estimate is divided by the expected value of this difference. That expected value is the square root of the sum of the specified observation error variance and the prior ensemble variance. If this ratio is greater than a threshold, the observation is not used. The threshold ratio used here is three which is commonly used for large tropospheric applications in DART."

5. Figure 5: The authors did a good job in effectively presenting the inversion performance in Figure 5, but did not discuss the 200 hPa results in the main text.

**Response:** Thank you for pointing this out! We added the following statement to Section 4.2 where Figure 5 (now Figure 6) is discussed:

"At 200 hPa, the spatial distribution of CO difference caused by assimilation is small in Experiment (2) Profile JNT assimilation, followed by Experiment (5) Profile TIR and column NIR assimilation. On the contrary, for the other three experiments which do not involve profile assimilations, the spatial distribution of CO difference caused by assimilation is relatively large. I.e., assimilating MOPITT profile product(s) only slightly change CO values at 200 hPa whereas assimilating MOPITT column product(s) changes CO values at 200 hPa dramatically. This is expected as vertical distribution is often an advantage of profile DA that column DA cannot represent."

6. Figure 9: The two assimilations with profile observations exhibit >1 line fit. Does this indicate assimilations with profile measurements tend to overestimate emissions or surface concentration?

**Response:** Even though the least square regression lines of the experiments involving profile assimilation seem to be above the 1:1 1ine while the least square regression lines of the experiments only involving column assimilation seem to be below the 1:1 line, the mean biases are positive for all the five experiments. This indicates that all five experiments, regardless of profile assimilation or column assimilation, tend to overestimate surface CO compared to CO observations from the NOAA Carbon Cycle Greenhouse Gases (CCGG) sites during July 31st, 2018 to August 14th. To avoid any confusion, we removed the least square regression lines from the figure. Nevertheless, as discussed in Section 6.1, the reviewer is right that the experiments involving profile assimilations tend to have higher values in emissions (Figure 13) and surface concentrations (Figure S15).

7. Line 175: Please clarify the approach to project MOPITT CO AK and prior to the model resolution.

**Response:** Instead of "In this case, the forward operators apply MOPITT averaging kernel and prior information to model CO field before comparing it to MOPITT products. The capability of assimilating MOPITT profile products is described in Barré et al., (2015)."

We rephrase to "We use the forward operators introduced in Barré et al., (2015), consisting of i) estimating the log of a pressure weighted partial column volume mixing ratio that corresponds to the MOPITT grid and ii) applying the MOPITT averaging kernel and prior information."

We also added the new Figure 1 to show MOPITT CO AK.

[Figure]

**Figure 1**. Averaging kernel (AK) rows for MOPITT retrieval types TIR only, NIR only, and multispectral TIR+NIR. Global average of AKs during July and August 2018 are shown.

8. Line 195 and 196: the order of case 1 and 2 are opposite in the text and Figure 2.

**Response:** Thank you. We have updated Figure 2 (now Figure 3).

**9. Line 211: Did the authors consider a spin-down timeframe? For observations after the assimilation timeframe while can still reflect emission signals?**

**Response:** Regarding initial conditions, the improvement will theoretically improve the CO distribution for as long as the CO lifetime, which varies in space at the global scale (~1 or 2 months). Regarding emissions, Gaubert et al. (2023) showed that using posterior emissions can improve CO for years-long simulations by mitigating systematic errors in emission fluxes.

**10. Line 320: Please clarify the definition of observation error variance, and if such error estimates have been incorporated into the assimilation system.**

**Response:** The observation error variance has been incorporated into the assimilation system. We are not aware of a data assimilation system that would not use observation error variance. We use the measurement error rather than total error provided by MOPITT for both column and profile products as smoothing error is already addressed by observation operators in the system. Specifically, for MOPITT profile products, measurement error is provided by the variable "MeasurementErrorCovarianceMatrix" while for MOPITT column products, measurement error is provided by the variable second column of the "RetrievedCOTotalColumnDiagnosticsDay". This information has been included in the revised manuscript; also see response to comment 3.

**11. Line 331: Please clarify the meaning of x in eq 4.**

**Response:** "x" on the right of the equation is the modeled state. χ on the left of the equation is a greek letter chi, the Chi-squared test has been introduced in statistics during the 19th century. In the context of data assimilation, it has been introduced in Menard and Chang (2000; https://journals.ametsoc.org/view/journals/mwre/128/8/1520-0493_2000_128_2672_aoscto_2.0.co_2.xml).

**12. Line 351: Experiment 2 and 5 are expected to assimilate similar information, with the major difference at 200 hPa, as indicated in Figure 5.**

**Response:** As explained in Section 2.4 (line 207), the comparisons of experiment (2) and (5) will show the impacts of assimilating joint products (TIR+NIR) versus assimilating them separately for profile products. The two different approaches will impact the vertical distribution. In the profile DA experiment, there is an attempt to provide more freedom to the vertical distribution, but it is localized to lower levels. The column assimilation impacts all vertical layers and does not

include the vertical localization. Therefore column assimilation modifies the upper troposphere and the stratosphere more.

13. Line 420: The mean bias in text is 0.7 ppb, while 5.72 in Figure 9. The correlation is 0.65 in text, while 0.79 in Figure 9.

**Response:** Thank you for noticing. The numbers in the Figure are correct. We have updated the numbers in the text.

14. Line 465: Given the assimilation timeframe is 15 days, please clarify the approach to estimate annual emissions.

**Response:** Note that the weather spin-up is 10 days. We then update the emissions for a total of 35 days (July 11 – August 15), the unit of Tg/year is only a unit conversion. We discuss the update of annual emissions in Gaubert et al. (2023).

15. Section 7.4 How about temporal resolution? Can MOPITT optimize emissions up to daily scale based on the experiments here?

**Response:** The prior fire emission is at daily temporal resolution. As in previous studies (Gaubert et al., 2020, 2023), the daily flux is updated, but the increment is also applied to future days, with an exponentially decreasing weight and set to zero at 16 days.

Technical corrections

16. Table 1: Missing the labels of TEMPO spectral ranges and potential chemical species for geostationary satellites.

**Response:** Thank you. We have added missing information. Please see below.

**Table 1**. Developed and potential multispectral satellite retrievals. Shown in the table are satellites, their NIR and/or TIR spectral ranges (in µm), and potential chemical species from the multispectral retrievals.

| Morning Overpass | Afternoon Overpass | Geostationary |
| --- | --- | --- |
| MOPITT (2.3 & 4.7) | AIRS (3.75–15.4) + OMI (0.27–0.5) | GIIRS (East Asia) (0.55–14.2) + TROPOMI (2.3–2.4) |
| (CO) | (O3) | (CO, O3) |
| IASI (3.6–15.5) + GOME2 (0.24–0.79) | TES (8.7–10.5) + OMI (0.27–0.5) | GEMS (East Asia) (0.3–0.5) + IASI (3.6–15.5) |
| (O3) | (O3) | (O3) |
| | GOSAT (0.75–15) + TES (8.7–10.5) | GEMS (East Asia) (0.3–0.5) + CrIS (3.9–15.4) |
| | (O3) | (O3) |

| | |
|---|---|
| CrIS (3.9–15.4) + GOSAT-2 (0.3–14.3)

(CO, CH4) | TEMPO (N. America) (0.29–0.74) + IASI (3.6–15.5)

(O3) |
| CrIS (3.9–15.4) + TROPOMI (2.3–2.4)

(CO, O3, CH4) | TEMPO (N. America) (0.29–0.74) + CrIS (3.9–15.4)

(O3) |

17. Line 130: typo, an extra ";" after "CAM-chem".

**Response:** Thank you. We have corrected it.

---

## Author Comment (AC2)

Referee #2
General Comments:

This study provides a comparative analysis by assimilating different MOPITT CO datasets to evaluate the advantages of assimilating multi-spectral satellite retrievals. This topic is important for better usage of MOPITT observations in the future. However, the manuscript focuses on the demonstration of assimilation results but lacks a further explanation for the possible causes of their large discrepancies, which reduces the importance of this manuscript. Furthermore, the content of this manuscript may need to be better organized. For example, the title of Section 4 is "Results". The audience may assume it is the major discussion section, however, it is followed by Sections 5-7. In addition, there are lots of typo errors. Overall, this is a good study with potentially important impacts on the application of satellite data. However, a significant improvement may need to be made before the paper can be considered for publication.

**Response:** Thank you. We agree with the reviewer and have re-organized the manuscript accordingly.

Specific Comments:

1. The authors raised three questions about the potential advantages of assimilating multispectral/joint retrievals. The developed and potential multispectral satellite retrievals are demonstrated in Table 1, including species of CO, O3, and CH4. More discussions are suggested to clarify the possible limitation of this analysis, i.e., whether the conclusion based on MOPITT CO is applicable to other instruments and species.

**Response:** As introduced in the introduction, regarding multispectral capabilities, MOPITT is a unique instrument as it retrieves total column amounts and vertical profiles of CO using both thermal-infrared (TIR) and near-infrared (NIR) measurements, and provides a multispectral TIR-NIR joint product. Therefore the MOPITT instrument is an ideal instrument to demonstrate the value of assimilating multi-spectral satellite retrievals of atmospheric composition and address these three questions. This study using MOPITT as a demonstration sheds light on the impacts of assimilating different satellite products of the same atmospheric composition. To address the reviewer's comment, we added the following statement to Section 7.4 where we discuss limitations:

"This study provides guidance for future work on the assimilation of multi-spectral satellite retrievals of atmospheric composition using MOPITT as a demonstration. However, whether the conclusions based on MOPITT CO are applicable to other species (e.g., CH4 and O3) needs further study."

2. The authors demonstrate dramatic discrepancies among different assimilations but lack a further explanation for the possible causes. For example, Figure 5: there are very large differences in the upper troposphere level. What are the possible causes and what are the possible impacts on the future assimilation of MOPITT CO data? Figure 10: Which assimilations have better performance in the upper troposphere, given their large discrepancies as shown in Figure 5? Section 6.1: The authors state "Assimilating profile products (Experiments (2) and (5)) tends to have a larger change to the emissions compared to only assimilating column products (Experiments (1), (3), and (4))". However, the change by assimilating Profile JNT data is very small, which cannot be described as "tends to have a larger change". The large discrepancy in Figure 13 by assimilating different datasets needs to be explained. Section 6.2: same as the above question, the change by assimilating Profile JNT data is almost ignorable, I don't understand why it could lead to better agreement with observations.

**Response:** Here we use ensembles to correct model priors with observed quantities. Localization is commonly used in ensemble-based data assimilation to address insufficient ensemble samples to ameliorate the spurious long-range correlations between the background and observations. In this study vertical localization is used. The main difference between profile assimilation and column assimilation is the involvement and treatment of the vertical information. For each MOPITT retrieval, profile products have multiple observations at different layers but their impacts are vertically localized around 100 hPa. Therefore not all vertical layers will be impacted. For the column, all vertical levels will be impacted by a single column value. In this case, if the mismatch is due to an underestimation of surface emissions rather than weak vertical transport, updating the upper tropospheric CO might lead to erroneous adjustments in CO abundance. In this study, the apparent contradiction between profile assimilation and column assimilation demonstrates the impacts of the inclusion and treatment of vertical information. Figure S4 shows the vertical profile increment for the experiments.

[Figure]

**Figure S4.** Vertical profile of the 15-day (July 31 - August 14, 2018) average increment in CO. Increment is calculated as analysis minus forecast.

We added more discussion in the manuscript. We added the following description of "vertical localization" in the manuscript. "*Localization is commonly used in ensemble-based data assimilation to address insufficient ensemble sample size. Since the correlation is expected to decrease as separation increases, it empirically reduces the impact of an observation on model state variable as a function of distance using the Gaspari–Cohn localization function. The spatial localization horizontal half width is 600 km and the vertical half width is 1200 m. The main difference between the profile and the column assimilation resides in the vertical localization. For each MOPITT retrieval, profile products have multiple observations at different layers but their impacts are vertically localized around 100 hPa. Therefore, not all vertical layers will be impacted. For the column data assimilation, there is no vertical localization in the column data assimilation except that the stratospheric (top 5) levels are not updated, as in the CO profile and meteorological DA. All vertical levels will be impacted by a single column value. In this case, if the signal is coming from an underestimation of surface emissions, correcting the upper troposphere might be wrong.*"

For Figure 5 (now Figure 6), we added "At 200 hPa, the spatial distribution of the CO difference caused by assimilation is smallest in Exp2-PJ, followed by Exp5-PT+CN. On the contrary, for the other three experiments which do not involve profile assimilations, the spatial distribution of the

CO difference caused by assimilation is relatively large. I.e., assimilating MOPITT profile product(s) only slightly changes CO values at 200 hPa whereas assimilating MOPITT column product(s) changes CO values at 200 hPa dramatically. This is expected as vertical distribution is often an advantage of profile DA that column can not represent."

For Figure 10 (now Figure 11), we added "Above 200 hPa, all five experiments overall agree better with IAGOS CO compared to the control run. However, experiments involving profile assimilation do not show obvious differences compared to experiments only involving column assimilation above 200 hPa."

For Section 6.1 and Section 6.2: The change by assimilating Profile JNT data is not small, and "tends to have a larger change" is the appropriate description as shown by Figure 12 (now Figure 13) (red line).

[Figure]

**Figure 13**. Updated (a-g) CAMS anthropogenic CO emissions and (h-n) FINNv2.4 fire CO emissions as a result of assimilating different MOPITT products. The emissions from the Spinup/control run are the unchanged original emissions of CAMS and FINNv2.4.

For Figure 13 (now Figure 14), we added "As shown previously, profile assimilation can out-perform column assimilations near the surface due to vertical localization. Different CO

concentrations at and near the surface resulted in different emission updates between profile assimilation and column assimilation."

3. There are lots of typo errors. It seems the manuscript was not checked carefully. For example, Lines 421-423: "(1) Column JNT assimilation, (2) Profile JNT assimilation, (3) Column TIR assimilation, (4) Column TIR and column NIR assimilation, and (5) Profile TIR and column NIR assimilation". Why are the five experiments listed here? Line 568 in the conclusion section, the authors state: "Results were not improved compared to WE-CAN because …" Because what?

**Response:** Thank you for pointing this out. We have carefully revised the manuscript to remove the typos.

Technical Comments:

4. Line 32: "vertical localization" is frequently used in the manuscript, but the meaning is unclear.

Response: We added the following description of "vertical localization" in the manuscript. "Localization is commonly used in ensemble-based data assimilation to address insufficient ensemble sample size. The error in the relation between an observation and a state variable is expected to increase as correlation between them decreases. Localization empirically reduces the impact of an observation on model state variable as a function of distance since correlation is expected to decrease as separation increases. The spatial localization horizontal half width is 600 km and the vertical half width is 1200 m. The main difference between the profile and the column assimilation resides in the vertical localization. For each MOPITT retrieval, profile products have multiple observations at different layers but their impacts are vertically localized around 100 hPa. Therefore not all vertical layers will be impacted. For the column data assimilation, there is no vertical localization in the column data assimilation except that the stratospheric (top 5) levels are not updated, as in the CO profile and meteorological DA. All vertical levels will be impacted by a single column value. In this case, if the mismatch is due to an underestimation of surface emissions rather than weak vertical transport, updating the upper tropospheric CO might lead to erroneous adjustments in CO abundance."

5. Table 1: what species are observed by the Geostationary satellites?

**Response:** We have updated Table 1. Please see below.

**Table 1**. Developed and potential multispectral satellite retrievals. Shown in the table are satellites, their NIR and/or TIR spectral ranges (in μm), and potential chemical species from the multispectral retrievals.

| Morning Overpass | Afternoon Overpass | Geostationary |
| --- | --- | --- |
| MOPITT (2.3 & 4.7) | AIRS (3.75–15.4) + OMI (0.27–0.5) | GIIRS (East Asia) (0.55–14.2) + TROPOMI (2.3–2.4) |
| (CO) | (O3) | (CO, O3) |
| IASI (3.6–15.5) + GOME2 (0.24–0.79) | TES (8.7–10.5) + OMI (0.27–0.5) | GEMS (East Asia) (0.3–0.5) + IASI (3.6–15.5) |
| (O3) | (O3) | (O3) |
| | GOSAT (0.75–15) + TES (8.7–10.5) | GEMS (East Asia) (0.3–0.5) + CrIS (3.9–15.4) |
| | (O3) | (O3) |
| | CrIS (3.9–15.4) + GOSAT-2 (0.3–14.3) | TEMPO (N. America) (0.29–0.74) + IASI (3.6–15.5) |
| | (CO, CH4) | (O3) |
| | CrIS (3.9–15.4) + TROPOMI (2.3–2.4) | TEMPO (N. America) (0.29–0.74) + CrIS (3.9–15.4) |
| | (CO, O3, CH4) | (O3) |

6. Lines 155-156: "DART assimilates observations and produce the analysis, an ensemble of optimized initial conditions"; then in lines 164-166, it seems the emissions are also updated. What are the actual objectives to optimize? While references are provided, it would be better if the authors could clarify the assimilation methodology more clearly to facilitate the readers.

**Response:** We have updated the following statement:

"To assimilate meteorology and chemical observational data, an ensemble of 30 CAM-chem simulations with different initial conditions and emissions to generate the forecast ensemble at a given time. DART assimilates observations and produce the analysis, an ensemble of optimized initial conditions (see details in Gaubert et al., 2016)."
 to

"Here, we use the Ensemble Adjustment Kalman Filter approach (EAKF; Anderson, 2001, 2003). The forecast ensemble is generated by 30 CAM-chem simulations with different initial conditions and emissions. The assimilation is performed using DART and produces an ensemble of optimized initial conditions and emissions, as described in Gaubert et al. (2023). Specifically the state vector includes CO initial conditions, and CO emission fluxes that are ascribed to fires and anthropogenic sources."

7. Line 210: typo error: "run" at the beginning of this line.

**Response:** This is not a typo. "The experiment runs start on July 16th 2018 and are initialized with the spin-up/control run."

8. Line 213: it could be better to explain the cause of the higher fraction at the beginning of the assimilation here.

**Response:** We added the following statement to Section 2.4:

"Systematic errors are larger at the beginning of the spinup, explaining the higher rejection rate. As the assimilation proceeds and the forecast bias is reduced, the rejection rate goes down."

9. The repeated usage of full experiment names, e.g., "Experiment (2) Profile JNT assimilation and Experiment (5) Profile TIR and column NIR assimilation" made the discussion hard to read. The authors may consider abbreviated names.

**Response:** Thank you for the suggestion. We have added the following abbreviated names and updated the manuscript accordingly. Please see the revised manuscript for details.

(1) Column JNT assimilation (Exp1-CJ);
(2) Profile JNT assimilation (Exp2-PJ);
(3) Column TIR assimilation (Exp3-CT);
(4) Column TIR and Column NIR assimilation (Exp4-CT+CN);
(5) Profile TIR and Column NIR assimilation (Exp5-PT+CN).

10. The Conclusion section needs better organization. It is not good to list 9 main findings in the Conclusion.

**Response:** We agree with the reviewer and rephrased the conclusions accordingly. Specifically, we removed a few conclusions that are less important and merged the rest to 5 main conclusions.